# A multiscale view of the Phanerozoic fossil record reveals the three major biotic transitions

Alexis Rojas [1✉], Joaquin Calatayud [1], Michał Kowalewski[2], Magnus Neuman [1] & Martin Rosvall [1]

The hypothesis of the Great Evolutionary Faunas is a foundational concept of macroevolutionary research postulating that three global mega-assemblages have dominated Phanerozoic oceans following abrupt biotic transitions. Empirical estimates of this large-scale pattern depend on several methodological decisions and are based on approaches unable to capture multiscale dynamics of the underlying Earth-Life System. Combining a multilayer network representation of fossil data with a multilevel clustering that eliminates the subjectivity inherent to distance-based approaches, we demonstrate that Phanerozoic oceans sequentially harbored four global benthic mega-assemblages. Shifts in dominance patterns among these global marine mega-assemblages were abrupt (end-Cambrian 494 Ma; end-Permian 252 Ma) or protracted (mid-Cretaceous 129 Ma), and represent the three major biotic transitions in Earth's history. Our findings suggest that gradual ecological changes associated with the Mesozoic Marine Revolution triggered a protracted biotic transition comparable in magnitude to the end-Permian transition initiated by the most severe biotic crisis of the past 500 million years. Overall, our study supports the notion that both long-term ecological changes and major geological events have played crucial roles in shaping the mega-assemblages that dominated Phanerozoic oceans.

[1] Integrated Science Lab, Department of Physics, Umeå University, Umeå 90736, Sweden. [2] Florida Museum of Natural History, Division of Invertebrate Paleontology, University of Florida, Gainesville 32611 FL, USA. ✉email: alexis.rojas-briceno@umu.se

Sepkoski's hypothesis of the Three Great Evolutionary Faunas that sequentially dominated Phanerozoic oceans represents one of the foundational concepts of macroevolutionary research. This hypothesis postulates that the major groups of marine animals archived in the Phanerozoic fossil record were non-randomly distributed through time and can be grouped into Cambrian, Paleozoic, and Modern evolutionary faunas[1]. Sepkoski formulated this three-phase model based on a factor analysis of family-level diversity[2], which became a framework-setting assumption in studies on the evolution of marine faunas and ecosystems[3–6], changing our view of the Phanerozoic history of life. However, while the hypothesis continues to serve as a conceptual platform for many recent studies[5,7–9], some analyses question the validity of Sepkoski's hypothesis[10]. Moreover, because Sepkoski's study predicts unusual volatility in the Modern evolutionary fauna starting during the mid-Cretaceous, a three-phase model fails to capture the overall diversity dynamics during long portions of the Mesozoic[11]. Whether such mid-Cretaceous radiation[12] represents an intra-faunal dynamic or a biotic transition from Sepkoski's Modern evolutionary fauna towards a neglected mid-Cretaceous-Cenozoic fauna remains unexplored.

It has been long recognized that Phanerozoic diversity is highly structured[13], provided the first global assessment based on a qualitative approach. However, empirical estimates of the macroevolutionary pattern depend on several methodological decisions, including background assumptions, statistical threshold, and hierarchical level[1,6,11]. The choice of input data also matter: Sepkoski's compendia and benthic taxa from the Paleobiology Database[1,14,15] generate different results. These limitations raise two fundamental questions: How can we identify global-scale mega-assemblage shifts without relying on critical methodological decisions? And given the underlying Earth-Life System, how should we represent the paleontological input data to accurately capture complex interdependencies? These limitations result in methodologically volatile and often inconsistent estimates of large-scale macroevolutionary structures, thus obscuring the causative drivers that underlie biotic transitions between successive global mega-assemblages. As a result, whether abrupt global perturbations, such as large bolide impacts or massive volcanic eruptions[16,17], and long-term ecological changes[18] both operate at the higher levels of the macroevolutionary hierarchy remains unclear[18,19].

Our understanding of the macroevolutionary dynamics of Phanerozoic life is being transformed by network-based approaches[6,20–22]. Because the input network can capture the complexity inherent to the underlying system, network analysis has become an increasingly popular alternative to the typical procedures used in almost every area of paleontological research[10,23–25]. However, as might be expected of an emergent interdisciplinary field, methodological inconsistencies and conceptual issues in network-based paleobiology research make it difficult to compare outcomes across studies. Also, the rapid development of the broader field of network science demands a major effort from paleobiologists working across disciplinary boundaries. Current network-based approaches in paleobiology[26] use standard network representations based on pairwise statistics that do not explicitly represent higher-order relationships[27] in the paleontological data. In addition, clustering methods are limited to a single scale of analysis and do not capture the multiscale dynamics[28] of the underlying Earth-Life System.

Here we implemented a multilayer network framework that captures higher-order temporal structures that emerge in multilayer networks describing temporal data[29]. In practice, we created an input network that explicitly represents different geological stages[30] as ordered layers assembled into a multilayer network. With multilevel hierarchical clustering[28], we tested for the major

biotic transitions in the Phanerozoic fossil record of the benthic marine faunas[22]. This multilayer network approach is transforming research on higher-order structures in both natural and social systems[31], and can help us to understand the metazoan macroevolution. We demonstrate that Phanerozoic oceans sequentially harbored four global mega-assemblages that scale up from lower-scale biogeographic structures and are defined by shifts in dominant faunas across the major biotic transitions in Earth's history. We found that abrupt global perturbations and long-term changes both played crucial roles in mega-assemblages transitions. Our study sheds light on the emergence of large-scale macroevolutionary structures[16,32]. For example, we show that biogeographic structures underlie the marine evolutionary faunas and that long-term changes controlled the shift to the modern mega-assemblage, which first emerged during the early Mesozoic but did not become dominant until the mid-Cretaceous. Our multilayer network approach provides an integrative framework of the metazoan macroevolution for future research.

## Results and discussion

**A multilayer network representation of the Earth-Life System.** We could have constructed a simple network representation of the fossil record by using physical nodes to represent its components, the fossil taxa and the geographic areas where they have been described, and links between them to indicate their relationships[25]. In recent years, similar standard network representations, including bipartite networks (Fig. 1A) and one-mode projections (Fig. 1B) that are based on pairwise relationships and capture first-order dependencies in the raw data, have been central to network-based macroevolutionary research[6,21,22,25,26]. However, these standard network representations of the fossil record ignore the time-constrained relationships in the underlying paleontological data[15] (Supplementary Data 1). Projecting standard bipartite network representations into unipartite networks[33] of taxa or geographic areas distorts even more information in the paleontological data (Fig. 1B). Consequently, clustering methods applied to standard network representations fail to capture the higher-order temporal structures that can emerge in multilayer networks describing temporal data[28,29].

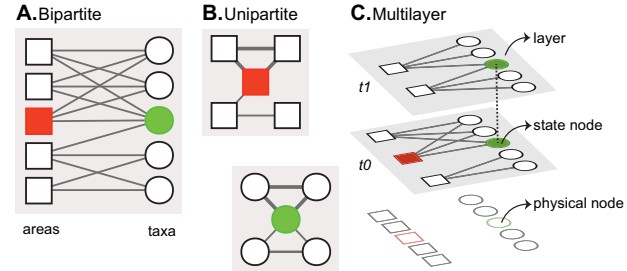

**Fig. 1 Network models used in macroevolution. A**, **B** Standard first-order network representations. **A** Bipartite occurrence network. This representation comprises two sets of physical nodes that represent geographic areas and taxa[25]. **B** Unipartite co-occurrence networks[6,22]. These representations are weighted projections of the bipartite network onto each set of physical nodes. **C** Higher-order multilayer representation of temporal data. In this network, distinct state nodes represent a physical node in each layer where the physical node occurs[28]. The trajectory of a random walker, guided by the links between the nodes, models network flows[38]. Network flows are of first order when random walker movements are constrained to single layers and of higher-order when they can move within and between layers[28]. The aggregation process that simplifies a multilayer network into a single-layer representation alters the network flows, obscuring the modular structures.

Multilayer network representations that consider higher-order dependencies in the row data[27,28], such as temporal relationships inherent to the fossil record known as the principle of faunal succession, have been shown to reveal community structures that standard models based on pairwise interactions cannot capture[34]. Clustering methods developed for standard networks cannot reliably identify overlapping modular structures[35]. Without the concept of layers, in the traditional network approaches used in macroevolutionary research, a random walker visiting a taxon will move at the same rate to grid cells from different ages because aggregating layers distorts temporal information (Fig. 1A, B). To overcome this limitation, we implemented a higher-order framework that explicitly represents different time intervals in Earth's history. In this network representation, a random walker visiting a physical node or taxon is located in a particular layer (t0) at a state node and will move at different rates to grid cells from the same layer and neighboring layers (Fig. 1C). In this way, we capture cross layer structures representing evolutionary faunas without destroying intralayer dynamics in the underlying system. Because the assembled network comprises nodes representing both animals and areas where they occur, we call our study system the Earth-Life System. Although we focus on the set of nodes representing animals, the bipartite network can also reveal spatiotemporal patterns in the distribution of both life and rocks.

Our multilayer network representation of the Earth-Life System treats ordered geological stages in the standard time-scale[30] as layers assembled into a multilayer network[27,36,37] (Supplementary Data 2). In this multilayer network, physical nodes, which in conventional models represent Phanerozoic benthic marine taxa[15] and geographic areas where they have been described, are divided into state nodes, which represent temporal relationships in the Earth-Life System (Fig. 1C). Specifically, for each taxon, we create one state node per layer where the taxon occurs. In this way, the trajectory of a random walker following the connections between state nodes in the multilayer network representation of stage-level fossil data captures higher-order temporal dependencies between the physical nodes. The trajectory of a random walker guided by the links between the nodes models network flows[38]. Network flows are of first order when random walker movements are constrained to individual layers and of higher order when they can move within and between layers[28]. Aggregation procedures that simplify a multilayer network representation of the Earth-Life System into a single-layer representation, such as those traditionally used in macroevolutionary research[6,26], change the trajectory of a random walker following the connections in the network, and inevitably obscure the macroevolutionary pattern (Supplementary Fig. 1).

We use the map equation multilayer framework[39], which operates directly on the assembled multilayer network and thereby preserves the higher-order interdependencies when identifying dynamical modular patterns in the underlying paleontological data. The map equation framework consists of an objective function that measures the quality of a given network partition[38], and an efficient search algorithm that optimizes this function over different solutions[28]. This algorithm provides the optimal multi-level solution for the input network, eliminating the subjectivity of distance-based approaches that provided different outputs at different thresholds[6,24]. Although our input network better represents the underlying Earth-Life System compared with standard network approaches[6,22,25], and our clustering approach allows to capture its optimal hierarchical modular structure, it can still be affected by numerous biases, including spatial and temporal variations in sampling effort, inequality in the rocks available for sampling, and taxonomic inconsistencies[40]. We generate and cluster bootstrap replicate networks to assess the impact of potential biases on the observed macroevolutionary pattern.

Specifically, we resampled taxon occurrences at each geographic cell from a Poisson distribution with mean equal to the number of occurrences per cell, and recalculated every link weight in the assembled multilayer network. Then, we clustered the bootstrap replicate networks with the same clustering approach we used for the assembled multilayer network (see "Robustness analysis").

In summary, our implementation of higher-order network models[28] to understand the benthic marine fossil record provides a more reliable coarse-grained description of the macroevolutionary patterns in two ways: first, we provide a more realistic input network that explicitly represents different time intervals in Earth's history and better captures the relationship in the underlying paleontological data. Second, we employ a clustering approach that operates directly on the multilayer network, delineates taxonomically overlapping assemblages, and automatically defines the number of assemblages or evolutionary faunas, which eliminates the subjectivity of the distance-based approaches. As a result, the macroevolutionary pattern revealed using multilayer networks integrates two major paleobiological hypotheses—the Mesozoic Marine Revolution and the Sepkoski's Great Evolutionary Faunas—into a single macroevolutionary story.

**The three major Phanerozoic biotic transitions**. We found that the assembled multilayer network is best described by four significant modules at the first hierarchical level, which correspond to Phanerozoic marine mega-assemblages of highly interconnected marine benthic taxa and geographic cells (Supplementary Data 3).

These large-scale modular structures characterize the underlying Earth-Life System (Fig. 2): The Phanerozoic oceans sequentially harbored four overlapping mega-assemblages defined by shifts in dominant faunas across the three major biotic transitions in Earth's history, taking place at end-Cambrian (~494 Ma), end-Permian (~452 Ma), and mid-Cretaceous (~129 Ma) times. The four-tier structuring of the Phanerozoic marine faunas differs from standard geological eras (adjusted mutual information, AMI = 0.71), indicating that not all major biotic transitions occur at their boundaries. Although different from the three units discriminated in Sepkoski's factor analysis[1], the classes of marine invertebrates that contribute the most to our Cambrian, Paleozoic, and combined Triassic–Cenozoic mega-assemblages match those from the hypothesis of the Three Great Evolutionary Faunas (Supplementary Fig. 2). This consensus suggests that the macroevolutionary units are unlikely to represent artifacts of the factor or network analyses.

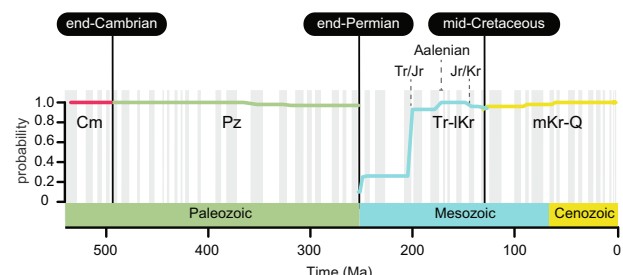

**Fig. 2 Phanerozoic oceans sequentially harbored four global benthic mega-assemblages.** The per stage significance measures the probability of retrieving a given mega-assemblage across 100 bootstrapped solutions and captures the global instability of the modular structure in the assembled network after the end-Permian[49] and subsequent Late Triassic and Early Jurassic extinction events[42]. Mega-assemblage shifts occur at the following boundaries: End-Cambrian (combined Paibian/Jiangshanian to Age10), end-Permian (Changhsingian to Induan), and mid-Cretaceous (Hauterivian to Barremian). Cm Cambrian, Pz Paleozoic, Tr-lKr Triassic to lower Cretaceous, mKr-Q mid-Cretaceous to Quaternary.

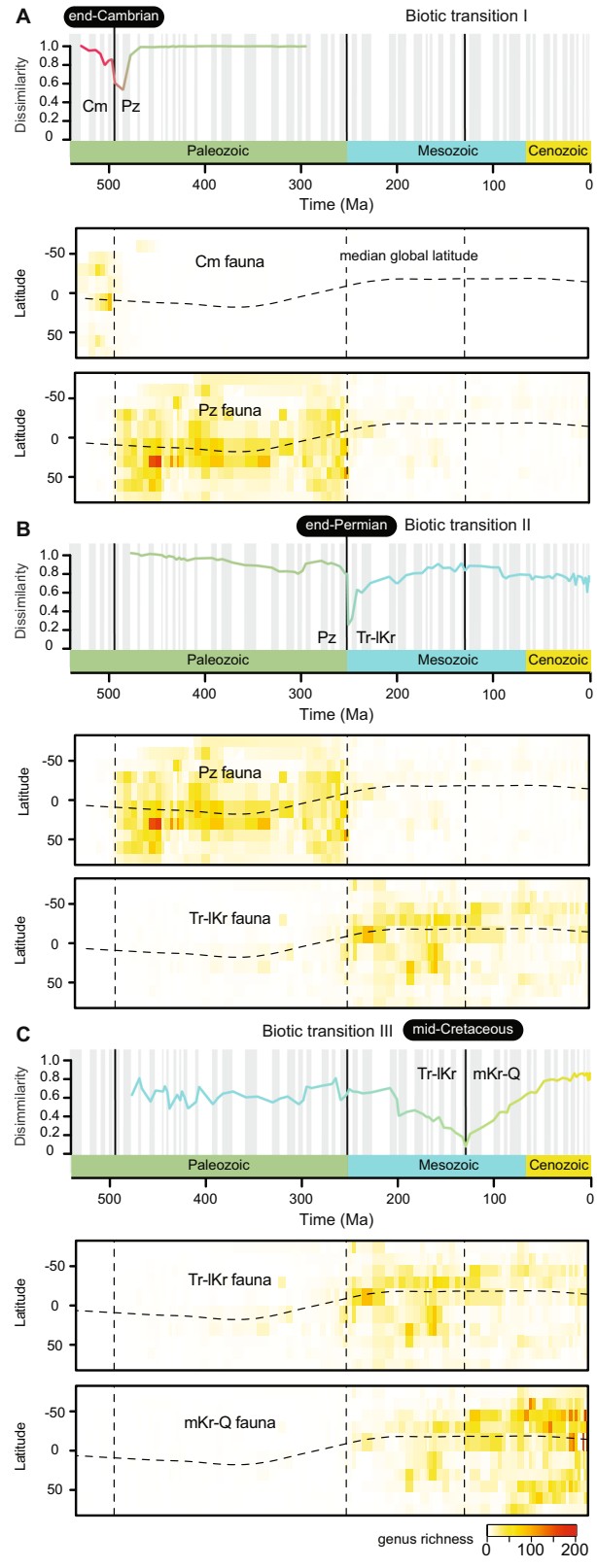

**Fig. 3 The three major biotic transitions in the Phanerozoic history of the benthic marine faunas. A** End-Cambrian. **B** End-Permian. **C** Mid-Cretaceous. The similarity of mega-assemblages at each biotic transition corresponds to the Sorensen dissimilarity $B_{sor}$[57]. Heatmaps of genus richness across time are interpolated from the underlying data for 10° latitudinal bands at each geological stage to indicate the latitudinal context. Shifts in dominance among global mega-assemblages are either abrupt global perturbations (biotic transitions I and II) or protracted changes (biotic transition III). Cm Cambrian, Pz Paleozoic, Tr-lKr Triassic to lower Cretaceous, mKr-Q mid-Cretaceous to Quaternary mega-assemblage.

The end-Permian mega-assemblage shift is also abrupt (Fig. 3B). The Paleozoic and Mesozoic mega-assemblages overlap in one geological stage and share only a few taxa (Jaccard similarity index = 0.03). This biotic transition coincides with the most severe biotic crisis of the past 500 million years[41], which is considered to have caused the global shift in ocean life at that time[42]. Therefore, the first two major biotic transitions delineated via multilayer network analysis were more likely triggered by mass extinction events. However, those global events did not governed subsequent evolutionary history of the marine biotas.

The mid-Cretaceous mega-assemblage shift is protracted, representing a gradual shift in dominance among two mega-assemblages that share more taxa (Jaccard similarity index = 0.11) and exhibit substantial overlap in geographic space (Fig. 3C). The protracted mid-Cretaceous mega-assemblage shift is reminiscent of the gradual Mesozoic restructuring of the global marine ecosystems, which included changes in food-web dynamics, functional ecology of dominant taxa, and increased predation pressure[12,43]. These changes in the marine ecosystems started during the Mesozoic and continued throughout the Cenozoic[44,45], but were particularly pronounced during the mid-Cretaceous[46]. Our results suggest that such changes in the global marine ecosystems may have been responsible for the gradual emergence of the modern benthic biotas. However, regardless of the specific transition mechanism, our results indicate that modern benthic biotas had already emerged during the early Mesozoic but did not become dominant until the mid-Cretaceous (~129 Ma).

In this way, the quadripartite structuring of the Phanerozoic marine fossil record captured by a multilayer network analysis couples the Three Great Evolutionary Faunas and the Mesozoic Marine Revolution hypothesis[1], which postulates the gradual diversification of the Modern evolutionary fauna during the Cretaceous[12]. Sepkoski's simulations arguably anticipated the Mesozoic transition[11] delineated in the multilayer network analysis presented here.

Alternative solutions reproduce either a four-phase model with a younger Cretaceous biotic transition, Sepkoski's three-phase model with biotic transitions occurring at era boundaries, or a three-phase model with a mid-Cretaceous but not end-Permian biotic transition. Regardless of the number of mega-assemblages delineated, these alternative solutions demonstrate that the major biotic transitions in Earth's history occurred across the end-Cambrian, end-Permian, and mid-Cretaceous boundaries. However, network clustering shows instability of the marine mega-assemblages at the geological stages following the Permian-Triassic boundary (Fig. 2). The significance of the mega-assemblages drops at this boundary and then increases, likely reflecting the recovery of the benthic marine faunas and ecosystems after the Earth's largest mass extinction event[47]. Although this punctuated recovery pattern should be further explored at finer temporal resolutions, our results suggest that full biotic recovery of the global biosphere from the end-Permian crisis was completed by the Early Jurassic. At this point, the Triassic-to-lower-Cretaceous mega-assemblage

The three major biotic transitions among the Phanerozoic marine mega-assemblages vary in timing and potential causative drivers. The end-Cambrian mega-assemblage shift appears to be an abrupt transition at the base of the uppermost Cambrian stage (Fig. 3A). However, the limited number of fossil occurrences from that interval precludes a better understanding of the transition.

(Tr-lKr) became robust: the probability of retrieving this globally dominant mega-assemblage across all bootstrapped solutions reaches high values (>0.9). Nevertheless, because both the Late Triassic and Early Jurassic extinctions should have contributed to the observed instability of the Tr-lKr mega-assemblage (Fig. 2), our results are consistent with claims[42,48,49] that complete ecological recovery from the Late Triassic and subsequent Early Jurassic crisis was attained by the Aalenian (Middle Jurassic). A mass extinction event initiated the biotic transition at the end-Permian, but more studies are required to understand the subsequent evolutionary history of the marine mega-assemblages.

Overall, our results support the suggestion that some global catastrophic events registered in the marine fossil record—including the end-Ordovician (Ashgillian), end-Devonian (Frasnian), and end-Cretaceous (Maastrichtian) mass extinctions, all considered within the major post-Cambrian biodiversity and ecological crises in Earth's history[42,50]—could have been less severe than previously proposed[41]. Regardless of the estimated magnitude of extinction, we demonstrate that the reduction in generic richness associated with such catastrophic events has little effect on the temporal arrangement of the global-scale evolutionary faunas (Fig. 4). Furthermore, our four-phase evolutionary pattern delineated via network analysis suggests that impacts of those extinction events on the marine biosphere were less severe than those from the protracted Mesozoic event that triggered the emergence of the re-delineated modern evolutionary fauna (the mid-Cretaceous to Quaternary mega-assemblage, mKr-Q).

**Implications for the macroevolutionary hierarchy.** We demonstrate that the Phaneozoic benthic marine faunas exhibit a hierarchically modularity in which first-level structures, representing the four Phanerozoic marine mega-assemblages, are built up from lower-level structures in a nested fashion (Fig. 5). The second-level structures underlying the four mega-assemblages represent sub-assemblages organized into time intervals that are equivalent to periods in the geological timescale (AMI = 0.83) (Supplementary Table 1). The third- and lower-level structures underlying the mega-assemblages form geographically coherent units[25] that change over geological time (Fig. 6). Likely due to limitations in the existing data, we were unable to map these evolutionary bioregions through the entire Phanerozoic. Nevertheless, our results demonstrate that local to regional biogeographic structures underlie the global-scale marine mega-assemblages in the macroevolutionary hierarchy. This multilevel organization of macroevolutionary units represents the large-scale spatiotemporal structure of the Phanerozoic marine diversity.

Without the inherent subjectivity of other approaches, our assessment of the Phanerozoic marine diversity, conducted simultaneously at different scales, can help us to comprehend the drivers and impacts most relevant at different macroevolutionary levels[16]. For instance, we show that both long-term ecological interactions and global geological perturbations seem to have played critical roles in shaping the mega-assemblages that dominated Phanerozoic oceans. However, some of the widely accepted major geological perturbations, including widely known global extinctions such as the Cretaceous–Paleogene event (K–Pg), control second-level but not first-level structures in this macroevolutionary hierarchy (Supplementary Table 1). Each level of organization in the emergent macroevolutionary hierarchy is structured as a network itself and can be studied independently (Fig. 5). Our integrative approach simultaneously quantifies both spatial and temporal aspects of the metazoan macroevolution and connects natural phenomena observed at global scales with those observed at local scales.

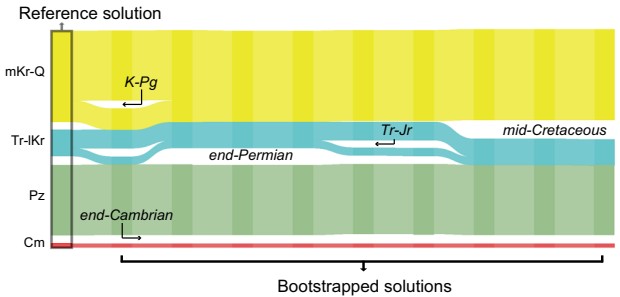

**Fig. 4 The solution landscape.** Alluvial diagram comparing the four-phase reference solution against alternative solutions obtained from bootstrapped networks. Vertical stacks connected by streamlines represent the networks. In each stack, a block represents nodes clustered into a module or mega-assemblage. Streamlines connect modules that contain the same nodes. The height of the streamline is proportional to the number of shared nodes between the connected modules. Splitting or merging streamlines indicate biotic transitions. The bootstrap strategy demonstrate high robustness of the four mega-assemblages in the reference solution (Supplementary Table 1), supporting the global biotic transitions delineated at the end-Cambrian, end-Permian, and mid-Cretaceous. The bootstrapped solutions also capture the small effect the Cretaceous–Paleogene extinction event (K–Pg) has on the mid-Cretaceous to Quaternary mega-assemblage (mKr-Q), and the instability of the global biosphere through the Triassic–Jurassic (Tr–Jr) extinction event (see Fig. 2). Cm Cambrian, Pz Paleozoic, Tr–lKr Triassic to lower Cretaceous, mKr–Q mid-Cretaceous to Quaternary mega-assemblages.

# Methods

**Data.** We used resolved genus-level occurrences derived from the Paleobiology Database (PaleoDB)[15], which at the time of access consisted of 79,976 fossil collections with 448,335 occurrences from 18,297 genera representing the well-preserved benthic marine invertebrates[22]. The PaleoDB assigns collections to paleogeographic coordinates based on their present-day geographic coordinates and age using GPlates[51]. We aggregated data using paleogeographic coordinates into a regular grid of hexagons covering the Earth's surface at each geological stage (4906 grid cells with count >0; inner diameter = 10° latitude–longitude) using the Hexbin R-package (http://github.com/edzer/hexbin). Aggregated fossil data are provided in the Supplementary Information (Supplementary Data 1). This binning procedure provides the symmetry of neighbors that is lacking in rectangular grids and captures the shape of geographic regions more naturally[52]. The selection of an optimum grid size is a compromise

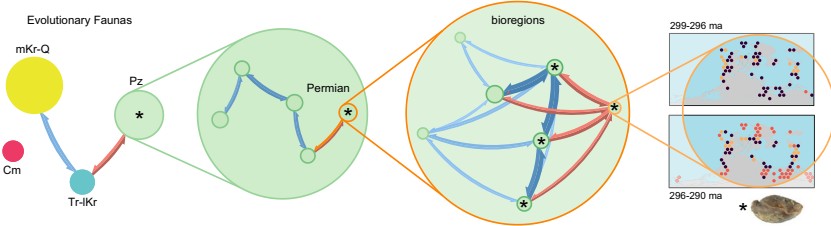

**Fig. 5 A macroevolutionary hierarchy for future research.** This visual representation shows the nested hierarchical structures in the reference solution (Supplementary Data 3). Modular structures at the higher level of organization in this macroevolutionary hierarchy correspond to evolutionary faunas, which are build up from lower levels entities, including sub-faunas, evolutionary bioregions, and taxa. Note that elements in this representation does not represent nodes from the original network but emergent structures and dynamics.

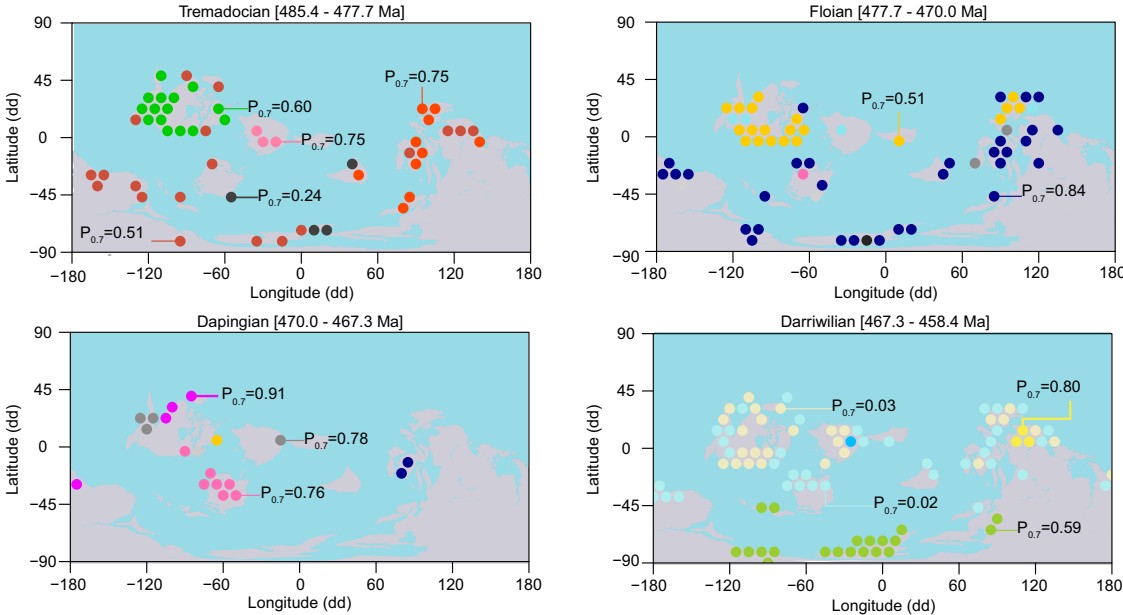

**Fig. 6 Examples of evolutionary bioregions recovered at lower hierarchical levels.** These modular structures form geographically coherent units that change through time and underlie the global marine mega-assemblages. Circles represent the center of the geographic cells colored by their module affiliation (Supplementary Data 3). By recovering bioregions at lower hierarchical levels, we demonstrate that our multilayer approach captures temporal (inter-stage) relationships without destroying spatial (intra-stage) dynamics.

between the lack of spatial resolution provided by hexagons with inner diameter = >10° and the increased number of hexagons without occurrences when shortening the inner diameter. However, recent studies have demonstrated that network analyses are robust to the shape (irregular, square, and hexagonal), size (5°–10° latitude–longitude), and coordinate system of the grid used to aggregate data[20,53].

**Network analysis**. We used the aggregated data to generate a weighted bipartite multilayer network[28], where layers represent ordered geological stages[30] and nodes represent taxa and geographic cells[25] (Fig. 1). We capture the collection-based structure of the underlying paleontological data[15] by joining taxa to geographic cells through weighted links ($w$). Specifically, for weight ($wki$) between geographic cell $k$ and taxa $i$, we divided the number of collections at grid cell $k$ that register taxa $i$ by the total number of collections recorded at geographic cell $k$. A similar link standardization has been employed in previous studies[22,25]. We combined the last two Cambrian stages, that is, Jiangshanian Stage (494–489.5 Ma) and Stage 10 (489.5–485.4 Ma), into a single layer to account for the lack of data from the younger Stage 10 and to maintain an ordered sequence in the multilayer network framework. However, our results show that the Cambrian to Paleozoic mega-assemblage shift occurred before the gap, and they are not directly related. The assembled network comprises 23,203 nodes ($n$), including 4906 spatiotemporal grid cells and 18,297 genera, joined by 144,754 links ($m$), distributed into 99 layers ($t$) (Supplementary Data 2).

We used the flow-based map equation multilayer framework with the search algorithm Infomap to cluster the assembled multilayer network[39]. This high-performance clustering approach[54] allowed us to model interlayer coupling based on the intralayer information of the multilayer network using a random walker[28]. The intralayer link structure represents the geographic constraints on network flows at a given geological stage in Earth's history, and the interlayer link structure represents the temporal ordering of those stages. In this neighborhood flow coupling, a random walker within a given layer moves between taxa and geographic cells guided by the weighted intralayer links with probability ($1-r$), and it moves guided by the weighted links in both the current and adjacent layers with a probability $r$. Consequently, the random walker tends to spend extended times in multilayer modules of strongly connected taxa and geographic cells that correspond to Phanerozoic marine mega-assemblages. Following the methodology of previous studies, we used the relax rate $r = 0.25$, which is large enough to enable interlayer temporal dependencies but small enough to preserve intralayer geographic information[55]. We tested the robustness to the selected relax rate by clustering the assembled network for a range of relax rates and compared each solution to the solution for $r = 0.25$ using Jaccard Similarity (Supplementary Fig. 3). We obtained the reference solution (Supplementary Data 3) using the assembled network and the following Infomap arguments: `-N 200 -i multilayer --multilayer-relax-rate 0.25 --multilayer-relax-limit 1`. The relax limit is the number of adjacent layers in each direction to which a random walker can move; a value of 1 enables temporal ordering of geological stages in the multilayer framework.

**Robustness analysis**. We employed a parametric bootstrap to estimate the significance of the four Phanerozoic mega-assemblages delineated in the reference solution. This standard approach accounts for the uncertainty in the weighted links connecting taxa to geographic cells due to numerous biases[40]. We resampled taxon occurrence at a given geographic cell using a truncated Poisson distribution with mean equal to the number of taxon occurrences. The truncated distribution has all probability mass between one and the total number of collections in the grid cell, thus avoiding false negatives. We obtained a resampled link weight for each link in the assembled network by dividing the sampled number by the total number of recorded collections in the grid cell. We included MATLAB code showing how we created bootstrap networks in the Supplementary Algorithm. Subsequently, we clustered the resulting bootstrap replicate network with the same clustering approach we used on the assembled network. We repeated this procedure—generating a bootstrap replicate network and clustering it into modules using Infomap—to generate 100 bootstrap modular descriptions. We compared the landscape of bootstrapped solutions against the reference solution to estimate the robustness of the four-tier macroevolutionary pattern. Specifically, for each reference module, we computed the proportion of bootstrapped solutions where we could find a module with Jaccard similarity higher than 0.5 (P05) and 0.7 (P07) (Supplementary Table 1 and Supplementary Data 4). In addition, we computed the average probability (median) of belonging to a module for nodes of the same layer (Fig. 2). This procedure for estimating module significance is described in ref. [56]. We visualized the modular descriptions with an alluvial diagram of several bootstrapped networks and the reference solution (Fig. 4).

**Statistics and reproducibility**. We clustered the networks into multilevel partitions with The Infomap Software Package[39]. To visualize and compare the multilevel network partitions, we used the Alluvial Generator and Network Navigator apps available on https://www.mapequation.org.

## Data availability
The underlying data are provided as Supplementary Data 1. The assembled multilayer network is provided as Supplementary Data 2. The reference solution is provided as Supplementary Data 3.

## Code availability
The code to generate bootstrap replicates of the assembled network is provided as Supplementary Algorithm in the Supplementary Information file.

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

## Acknowledgements
We thank the contributors to the Paleobiology Database who collected data. We thank S. Finnegan and D. Edler for useful discussions, and R. Nawrot for helpful comments on an early version of the manuscript. A.R. was supported by the Olle Engkvist Byggmästare Foundation, J.C. by the Carl Trygger Foundation, and M.R. by the Swedish Research Council, grant 2016-00796.

## Author contributions
A.R. conceived the project and performed the network analysis. A.R. and M.R. designed the experiments. J.C., A.R. and M.N. performed the robustness assessment. A.R., M.K. and M.R. wrote the manuscript with input from all authors. All authors discussed the results and commented on the manuscript.

## Funding

## Competing interests

The authors declare no competing interests.
