## [Peer Review File · Communications Biology]

Reviewers' comments:

Reviewer #1 (Remarks to the Author):

Review of "A multiscale view of the Phanerozoic fossil record reveals the three major biotic transitions"

First of all, apologies for the lateness of my review. Covid, working from home, lack of school/childcare, transition to online learning, and the recent exams fiasco in the UK have hit hard over the past few months.

I really enjoyed reading this paper. It is a very elegant approach to testing something that is often taken as palaeontological gospel whereas we all suspected that it might have been an oversimplification. However, reassuringly, the results aren't too different to what Sepkoski originally reported, with major, abrupt transitions between the Cambrian-Palaeozoic and Palaeozoic-"modern" faunas proving to be robust. What I really like about these results is that it gives a place to the Mesozoic Marine Revolution (MMR) in the story of Phanerozoic marine macroevolution. I have little to criticise in the manuscript as it is very well written and has a good narrative. However, there are two main points I feel need addressing and a minor few comments here and there. My main criticisms are that:

(i) In the spirit of facilitating reproducibility, the authors should outline the PBDB download procedure in more detail and submit the code used for the analyses. Both of these can easily just be added to the supplement. At present, I could not reproduce this study from what is given to me in the manuscript and supplement.

(ii) There isn't a figure that clearly shows the 2x abrupt changes and the protracted change across the late Mesozoic. I don't really know how easy or even possible this would be due to the data, but some sort of similarity index change through the Phanerozoic would be a really nice visual. Without this, it is difficult for someone not deeply versed in network methods to assess the significance of your transitions and mega-assemblages, i.e. there is nowhere in the main manuscript that highlights this, other than the description and discussion in the text. I feel this is a major downside to the current presentation of the study and needs addressing if the paper is going to make a significant impact in the field.

However, given my general lack of criticism and positive outlook on this paper, I recommend publication with minor revisions so long as the two main points above can be addressed. Thanks for sending me this to review and, again, apologies for the lateness of my response. I look forward to updating my macroevolution lecture once this paper is published!

Detailed comments below:

Line 141: more detail on your sampling bias methods, please. I suspect some of these analyses, particularly back into the Palaeozoic, will really be influenced by spatial sampling regime.

Line 229: Might be worth mentioning that the Late Triassic and subsequent Early Jurassic (i.e. Toarcian) extinctions would also have a hand in this instability. Dunhill et al. (2018) suggest an ultimate Aalenian (i.e. Middle Jurassic) full ecological recovery from the Late Triassic and subsequent Early Jurassic events. You do actually see ecological recovery between the two events in the Sinemurian but then the Early Jurassic event sets the system back again. This is because of delayed reef recovery in the tropics whereas level-bottom benthic recovery at higher latitudes appears to have been much faster (Hautmann et al. 2008).

Line 233: can reference Song et al. (2018) here too (already referenced as (32)) which suggests a Late Triassic culmination of full ecological recovery.

Line 261: I haven't seen the Cretaceous-Paleogene (K-Pg) extinction referred to as the Cretaceous-Tertiary for a number of years now. I'd go with K-Pg as it is consistent with the current accepted chronostratigraphic nomenclature.

Figure 2A. I really like this plot and I think it does an amazing job of highlighting the effect of the Permo-Triassic mass extinction and the inherent instability of the biosphere during the entire Triassic. This fits with the conclusions of our recent Sci Adv paper (Song et al. 2018) which you already reference, which highlights that ecological stability remained up until the very Late Triassic (at least).

Figure 2B. Is this just to show where the data comes from, in terms of latitudinal richness? The mention of abrupt and protracted changes is somewhat confusing in the legend because I can't identify them from the plot.

References

- Dunhill AM, Foster WJ, Sciberras J, and Twitchett RJ. 2018. Impact of the Late Triassic mass extinction on functional diversity and composition of marine ecosystems. *Palaeontology* 61:133-148.
- Hautmann, M. Stiller, F., Huawei, C. And J Ingeng, S. 2008. Extinction-recovery pattern of level-bottom faunas across the Triassic-Jurassic boundary in Tibet: implications for potential killing mechanisms. *Palaios*, 23, 711-718.

Best wishes,
Alex Dunhill
University of Leeds, UK

Reviewer #2 (Remarks to the Author):

First a personal caveat. I do not understand the methods used in this paper, so I am unable to comment on their validity or limitations. But if the authors are right, the conclusions they reach seem a reasonable, if highly abstract, description of Phanerozoic benthic marine transitions. Perhaps space limitations prevent a more thorough dissection of this pattern. Stanley and others have previously commented that some mass extinctions (most by the present account) did not have profound effects on marine faunas; but it would be nice to point this out. The authors cite the MMR as a symptom of the final transition, so it would be interesting to ask why previous episodes of gradual change, such as during the Silurian to Carboniferous, did not result in a similar gradual transition. One wonders if recognition of the mid-Cretaceous transition is somehow related to better sampling and simply more information than is available for earlier periods. There is also no attempt here to discuss what agencies propelled these transitions; again this is understandable because of space limitations, but again it would be nice to say at least something in a paragraph at the end. Sepkoski suffered from similar limitations, and perhaps for this reason I have never been a fan of the Three Evolutionary Faunas paradigm. The present paper may offer a more nuanced description but perhaps not much more clarity to the pattern. -Geerat J. Vermeij

Reviewer #3 (Remarks to the Author):

The manuscript titled "A multiscale view of the Phanerozoic fossil record reveals the three major biotic transitions" presents a reanalysis of a long-standing paradigm in Palaeontology, where the biota can be divided into 3 main evolutionary biotas. Furthermore, they present a novel suite of methods, at least for the field of palaeontology, which is "multilayer network framework". This gives the manuscript two potentially citable aspects and a lot of interest that makes it suitable for publication in Biology Communications. I do, however, have some concerns and comments. At this stage, I am concerned at the high similarity to a recently published article by Drew Muscente et al. PNAS 115, 5217-5222 (2018) which has since received 21 citations [excl. self citations]. Both articles use the same dataset, both use network analysis (even though the application here is more advanced), and both investigate changes in the evolutionary faunas. The results vary slightly with Muscente et al. recognizing 3-5 evolutionary faunas, and here only 3 and an earlier transition in the Cretaceous. I think others will also ask the same question; how does this new manuscript represent a significant advance? So, the follow up to that is, okay is this "multilayer network framework" the significant advance? After reading the manuscript I do not know, because I am struggling to understand what this multilayer approach is doing/ how it works – and that is also a concern because I would argue I have a head start over most having used and published a network analysis application in Palaeontology (see Scientific Reports 10, 2176 (2020)). To summarise, I like this research and it is an outstanding achievement to understand and apply these complicated new methods and it should, therefore, be published but before that I think it needs some changes in how the ideas are communicated so it is more understandable and so the advances to the field are also more understandable.

Also note, the dataset was not made available to me so fundamentally I could not review the data that underpins this study, and it appears that there was no vetting of the data. Stating "the dataset used in this study is freely available at the Paleobiology Database" is not acceptable because that database is always evolving, it is also standard practice to include the .csv file of the actual dataset that was used (see Muscente et al. as an example). Without including the data this study is not reproducible. Likewise, the code was not available so this study could not be reproduced – I recommend using mybinder.org to make all this accessible and reproducible.

Here are some more specific comments on the manuscript:

John Phillips 1860 (not Jack Sepkoski) was the first to estimate Phanerozoic diversity and divide the curve into evolutionary faunas dubbed the Palaeozoic, Mesozoic and Cenozoic Life based on the British fossil record. Sepkoski and Raup much later (> 100 years) were some of the first, however, to quantify these changes and came up with slightly different results. This highlights a long standing concern I have about research conducted in the US, and that is that typically a lot of work done elsewhere in the world is overlooked or looked down at (personally I think this is because of the way palaeontologists network and the consequences of conferences). There are other examples of this, e.g., Line 179: Penn et al. is not an appropriate reference for the sentence. P.S., your results are very similar to Phillips (1860) – which raises another point about statistical methods, where you should always use the simplest methods possible.

Multilayer network framework:

Line 61-21 "integrates the higher-order relationships over time" this does not mean anything; please can you be more specific. As someone who incorporates machine learning into analysing data, I would think that you mean you are using all the metadata to find relationships, which is not correct. I could also interpret this to mean that you are looking at different taxonomic levels, which is also not correct. As I understand the "layers" represent different time intervals, which would stop 2 nodes connecting if they're separated by a significant number of layers – if that is the case, how do you end up with fewer

evolutionary faunas then using a bipartite or unipartite method because this approach should be splitting the data more often?

You also explain the geographic range is used in the analysis, although I do not understand how, but this would then make the dataset prone to many sampling biases. I know you acknowledge this, but how does a parametric bootstrap method fix a lack of data/ incorrect data in the Paleobiology Database. You do not say in the actual article that you vetted the data. Furthermore, why does geography matter when studying evolutionary faunas? Does it matter that a family of snails that evolved in the tropics can now only be found in the depths of Antarctica when understanding evolutionary faunas?

Terms like "physical nodes" "state nodes" need explaining in a way an undergraduate would understand.

Lines 103-109. It is better to use well-known stages, right?
Like Maastrichtian.

Lines 125-143 do not explain what you are doing in a way that can be repeated without extensive additional work and reading.

You do not explain what data you are using, okay you say Paleobiology Database, but you also say taxa, is this species-level, genus-level, family-level, order-level? One aspect I like in Muscente et al. is that they run their analysis at all these levels to see the differences in the results.

Finally, I looked at ref 24 which you cite a lot. I am now asking the question why didn't you use a multilayer memory network? This would have been better adapted to handling sampling problems, especially problems like Lazarus taxa if I understood it right.

Three major biotic transitions

Lines 155-158 "differs from standard geological eras (Adjusted Mutual Information, AMI = 0.71), indicating that not all major biotic transitions occur at their boundaries." This is very hard to believe, biostratigraphy is based on major turnovers in fauna and accordingly this analysis did not [some would argue failed to] recognise the turnover at the Mesozoic/Cenozoic boundary. Since I could not see the data I could not go into further detail as to why this is the case, but isn't it strange that ammonites (I am using an extreme example) are clustering with Cenozoic genera? The issue is probably that you are only looking at invertebrates, if you included the chondrichthyans I am sure you would see a Cenozoic cluster appear. Another way to look at it, is does this multilayer method really work? If you did the bipartite and unipartite methods do you get different results?

My paper I mentioned before, *Scientific Reports* 10, 2176 (2020), did find a significant turnover in benthic invertebrates at the K/Pg boundary. It is probably then a good idea to compare this new method with the 'old' network methods you introduce earlier on. P.S., my issue with network analyses is that it seems to under cluster assemblages and this new approach does not appear to be immune to this problem.

Line 178-179: neither ref. 34 nor 35 are appropriate. You need citations that show evolutionary changes and comparisons of mass extinction events, these two papers do not do that and gives the appearance of a lack of understanding. I highlighted these ones because I know them best, so please check the quality of the referencing throughout.

Figure 5 is not cited in the manuscript.

I do not understand how Figs. 4-5 contribute to the article when you do not discuss these "lower

order" structures.

I do not understand Fig. 3 and why some of the bootstraps are highlighted as Sepkoski's evolutionary faunas – how would I know from the figure? What do the white gaps mean?

The formatting of the article was slightly off, means words had spaces within them making it harder to read. E.g., line 87 "network" "including". E.g., line 99.

Lines 14-22, these sentences are real tongue twisters and hard to read.

That written quality of the article was high!

I hope you have found my review constructive and I am available if you have any further questions,

Kind Regards
William Foster

Point-by-point response to the referees' comments

Response to Referee 1

I really enjoyed reading this paper. It is a very elegant approach to testing something that is often taken as palaeontological gospel whereas we all suspected that it might have been an oversimplification. However, reassuringly, the results aren't too different to what Sepkoski originally reported, with major, abrupt transitions between the Cambrian-Palaeozoic and Palaeozoic-modern faunas proving to be robust. What I really like about these results is that it gives a place to the Mesozoic Marine Revolution (MMR) in the story of Phanerozoic marine macroevolution. I have little to criticise in the manuscript as it is very well written and has a good narrative. However, there are two main points I feel need addressing and a minor few comments here and there.

We are thankful for the time and effort invested in reviewing our manuscript. We also sincerely appreciate the positive evaluation.

My main criticisms are that:

(i) In the spirit of facilitating reproducibility, the authors should outline the PBDB download procedure in more detail and submit the code used for the analyses. Both of these can easily just be added to the supplement. At present, I could not reproduce this study from what is given to me in the manuscript and supplement.

To facilitate reproducibility in the revised version, we have:

- Described the PBDB download procedure in more detail (Line 347).
- Included the PBDB filtered data (Data S1).
- Included the code to obtain bootstrapped networks from the original network (Supplementary Information).

In addition, we have indicated that the following resources are available online at the Infomap website www.mapequation.org:

- Source code for multilevel clustering with Infomap. It includes a client-side web application that makes it possible for users to run Infomap without any installation.
- Source code to perform the significance clustering.
- Source code to explore the landscape of solutions discussed in the appendix.
- The Alluvial Generator App we used to create Fig. 4.
- The Network Navigator App to create interactive zoomable maps for networks clustered with Infomap that we used used to generate Fig. 5.

(ii) There isnt a figure that clearly shows the 2x abrupt changes and the protracted change across the late Mesozoic. I dont really know how easy or even possible this would be due to the data, but some sort of similarity index change through

the Phanerozoic would be a really nice visual. Without this, it is difficult for someone not deeply versed in network methods to assess the significance of your transitions and mega-assemblages, i.e. there is nowhere in the main manuscript that highlights this, other than the description and discussion in the text. I feel this is a major downside to the current presentation of the study and needs addressing if the paper is going to make a significant impact in the field.

REVISIONS. This is a good point. In the revised version, we have quantified dissimilarity with the Sørensen dissimilarity index between the two consecutive faunas across each biotic transition, and added a visualization of the temporal changes. This figure highlights differences between abrupt transitions (biotic transitions I and II) and protracted transition (biotic transition III):

Figure 3. The three major biotic transitions in the Phanerozoic history of the benthic marine faunas. A. End-Cambrian. B. End-Permian. C. Mid-Cretaceous. The similarity of mega-assemblages at each biotic transition corresponds to the Sørensen dissimilarity (B_{sor}) (Baselga 2012). Heatmaps of genus richness across time are interpolated from the underlying data for 10° latitudinal bands at each geological stage only to indicate the latitudinal context. Shifts in dominance among global mega-assemblages are either abrupt global perturbations (biotic transitions I and II) or protracted changes (biotic transition III). Abbreviations: Cambrian (Cm); Paleozoic (Pz); Triassic to lower Cretaceous (Tr-IKr); and mid-Cretaceous to Quaternary mega-assemblage (mKr-Q).

However, given my general lack of criticism and positive outlook on this paper, I recommend publication with minor revisions so long as the two main points above can be addressed. Thanks for sending me this to review and, again, apologies for the lateness of my response. I look forward to updating my macroevolution lecture once this paper is published!

We sincerely appreciate this positive evaluation. We strongly believe that complex networks will revolutionize macroevolutionary research.

Detailed comments below:

Line 141: more detail on your sampling bias methods, please. I suspect some of these analyses, particularly back into the Palaeozoic, will really be influenced by spatial sampling regime.

REVISION. We have described the parametric bootstrap employed in our analysis in more detail. The revised paragraph in the Robustness analysis sub-section (Line 421):

We employed a parametric bootstrap to estimate the significance of the four Phanerozoic mega-assemblages delineated in the reference solution. This standard approach accounts for the uncertainty in the weighted links connecting taxa to geographic cells due to numerous biases Smith [2007]. We resampled taxon occurrence at a given geographic cell using a truncated Poisson distribution with mean equal to the number of taxon occurrences. The truncated distribution has all probability mass between one and the total number of collections in the grid cell, thus avoiding false negatives. We obtained a resampled link weight for each link in the assembled network by dividing the sampled number by the total number of recorded collections in the grid cell. Subsequently, we clustered the resulting bootstrap replicate network with the same clustering approach we used on the assembled network. We repeated this procedure – generating a bootstrap replicate network and clustering it into modules using Infomap – to generate 100 bootstrap mod-

ular descriptions (see supplementary materials). We compared the landscape of bootstrapped solutions against the reference solution to estimate the robustness of the four-tier macroevolutionary pattern. Specifically, for each reference module, we computed the proportion of bootstrapped solutions where we could find a module with Jaccard similarity higher than 0.5 (P05) and 0.7 (P07) (Tables S1-S2). In addition, we computed the average probability (median) of belonging to a supermodule for nodes of the same layer (Fig. A). This procedure for estimating module significance is described in ref. Calatayud et al. [2019]. We visualized the modular descriptions with an alluvial diagram of several bootstrapped networks and the reference solution.

REVISION. In the revised version, we also edited the paragraph that introduced our robustness analysis in the document for the first time (Line 141):

We generate and cluster bootstrap replicate networks to assess the impact of potential biases on the observed macroevolutionary pattern. Specifically, we resampled taxon occurrences at each geographic cell from a Poisson distribution with mean equal to the number of occurrences per cell, and recalculated every link weight in the assembled multilayer network. Then, we clustered the bootstrap replicate networks with the same clustering approach we used for the assembled multilayer network (see Robustness analysis).

Line 229: Might be worth mentioning that the Late Triassic and subsequent Early Jurassic (i.e. Toarcian) extinctions would also have a hand in this instability. Dunhill et al. (2018) suggest an ultimate Aalenian (i.e. Middle Jurassic) full ecological recovery from the Late Triassic and subsequent Early Jurassic events. You do actually see ecological recovery between the two events in the Sinemurian but then the Early Jurassic event sets the system back again. This is because of delayed reef recovery in the tropics whereas level-bottom benthic recovery at higher latitudes appears to have been much faster (Hautmann et al. 2008).

RESPONSE. We thank the reviewer for pointing us toward those useful references.

REVISION. We have revised the paragraph as requested and added the references (Line 278):

Although this punctuated recovery pattern should be further explored at finer temporal resolutions, our results suggest that full biotic recovery of the global biosphere from the end-Permian crisis was completed by the Early Jurassic. At this point, the Triassic-to-lower-Cretaceous mega-assemblage (Tr-lKr) became robust: the probability of retrieving this globally dominant mega-assemblage across all bootstrapped solutions reaches high values (>0.9). Nevertheless, because both the Late Triassic and Early Jurassic extinctions should have contributed to the observed instability of the Tr-lKr mega-assemblage (Fig. 2A), our results are consistent with an ultimate Middle Jurassic (Aalenian) complete ecological recovery from the Late Triassic and subsequent Early Jurassic crises Chen and Benton [2012], Song et al. [2018], Dunhill et al. [2018].

Overall, our results support the suggestion that some global catastrophic events registered in the marine fossil record – including the end-Ordovician (Ashgillian), end-Devonian (Frasnian), and end-Cretaceous (Maastrichtian) mass extinctions, all considered within the major post-Cambrian biodiversity and ecological crises in Earth’s history Hautmann et al. [2008], Dunhill et al. [2018] – could have been less severe than previously proposed Stanley [2016]. Regardless of the estimated magnitude of extinction, we demonstrate that the reduction in generic richness associated with such catastrophic events has little effect on the temporal arrangement of the global-scale evolutionary faunas (Figs. 1, 4). Furthermore, our four-phase evolutionary pattern delineated via network analysis suggests that their impacts on the marine biosphere were less severe than those from the protracted Mesozoic event that triggered the emergence of the re-delineated modern evolutionary fauna (the mid-Cretaceous to Quaternary mega-assemblage, mKr-Q).

REVISION. In the new figure 2, formerly 2A, we relabeled to help the

reader follow the discussion:

Figure 2. Phanerozoic oceans sequentially harboured four global benthic mega-assemblages. The per stage significance measures the probability of retrieving a given mega-assemblage across 100 bootstrapped solutions and captures the global instability of the modular structure in the assembled network after the end-Permian Song et al. [2018] and subsequent Late Triassic and Early Jurassic extinction events Dunhill et al. [2018]. Mega-assemblage shifts occur at the following boundaries: End-Cambrian (combined Paibian/Jiangshanian to Age10), end-Permian (Changhsingian to Induan), and mid-Cretaceous (Hauterivian to Barremian). Abbreviations: Cambrian (Cm); Paleozoic (Pz); Triassic to lower Cretaceous (Tr-lKr); and mid-Cretaceous to Quaternary (mKr-Q).

Line 233: can reference Song et al. (2018) here too (already referenced as (32)) which suggests a Late Triassic culmination of full ecological recovery.

REVISION. Revised as requested.

Line 261: I havent seen the Cretaceous-Paleogene (K-Pg) extinction referred to as the Cretaceous-Tertiary for a number of years now. Id go with K-Pg as it is consistent with the current accepted chronostratigraphic nomenclature.

REVISION: Revised as requested.

Figure 2A. I really like this plot and I think it does an amazing job of highlighting the effect of the Permo-Triassic mass extinction and the inherent instability of the biosphere during the entire Triassic. This fits with the conclusions of our recent Sci Adv paper (Song et al. 2018) which you already reference, which highlights that ecological stability remained up until the very Late Triassic (at least).
Figure 2B. Is this just to show where the data comes from, in terms of latitudinal richness? The mention of abrupt and protracted changes is somewhat confusing in the legend because I cant identify them from the plot.

RESPONSE. The reviewer is right. We made these figure panels to show where the data come from in terms of latitudinal richness. However, our paper is focused on the temporal patterns.

REVISION. In the revised version, we added a visualization of temporal changes in dissimilarity between the two consecutive faunas across each biotic transition. This new figure highlights differences between protracted and abrupt transitions (see Figure 3).

References

Dunhill AM, Foster WJ, Sciberras J, and Twitchett RJ. 2018. Impact of the Late Triassic mass extinction on functional diversity and composition of marine ecosystems. *Palaeontology* 61:133-148.

Hautmann, M. Stiller, F., Huawei, C. And J Ingeng, S. 2008. Extinction-recovery pattern of level-bottom faunas across the Triassic-Jurassic boundary in Tibet: implications for potential killing mechanisms. *Palaios*, 23, 711718.

REVISION. We have added both references.

Response to Referee 2

First a personal caveat. I do not understand the methods used in this paper, so I am unable to comment on their validity or limitations. But if the authors are right, the conclusions they reach seem a reasonable, if highly abstract, description of Phanerozoic benthic marine transitions. Perhaps space limitations prevent a more thorough dissection of this pattern. Stanley and others have previously commented that some mass extinctions (most by the present account) did not have profound effects on marine faunas; but it would be nice to point this out.

REVISION. We appreciate the reviewer's comments. In the revised version, we have added a paragraph pointing out that our results indicate that some mass extinction did not have profound effects on marine faunas. We have also included some references that have previously commented on this (Line 292):

Overall, our results support the suggestion that some global catastrophic events registered in the marine fossil record – including the end-Ordovician (Ashgillian), end-Devonian (Frasnian), and end-Cretaceous (Maastrichtian) mass extinctions, all considered within the major post-Cambrian biodiversity and ecological crises in Earth's history Hautmann et al. [2008], Dunhill et al. [2018] – could have been less severe than previously proposed Stanley

[2016]. Regardless of the estimated magnitude of extinction, we demonstrate that the reduction in generic richness associated with such catastrophic events has little effect on the temporal arrangement of the global-scale evolutionary faunas (Figs. 1, 4). Furthermore, our four-phase evolutionary pattern delineated via network analysis suggests that their impacts on the marine biosphere were less severe than those from the protracted Mesozoic event that triggered the emergence of the re-delineated modern evolutionary fauna (the mid-Cretaceous to Quaternary mega-assemblage, mKr-Q).

The authors cite the MMR as a symptom of the final transition, so it would be interesting to ask why previous episodes of gradual change, such as during the Silurian to Carboniferous, did not result in a similar gradual transition. One wonders if recognition of the mid-Cretaceous transition is somehow related to better sampling and simply more information than is available for earlier periods.

REVISION: This idea is interesting. Nevertheless, we showed that changes in generic richness have little effect on the temporal changes in dominance patterns of the global-scale evolutionary faunas. That explains why the major biotic transitions delineated in our study does not match all the known mass extinctions events. In contrast, biotic transitions emerge from changes in connectivity between adjacent layers (faunal replacements between adjacent geological stages), regardless of potential variations in genus richness. To clarify this point, we follow the indications by reviewers 1 and 2. In the revised version, we have added the paragraph (see above) highlighting that our results indicate that some mass extinction did not have profound effects on marine faunas. We also included some references that have previously commented on this (Line 292):

There is also no attempt here to discuss what agencies propelled these transitions; again this is understandable because of space limitations, but again it would be nice to say at least something in a paragraph at the end. Sepkoski suffered from similar limitations, and perhaps for this

reason I have never been a fan of the Three Evolutionary Faunas paradigm. The present paper may offer a more nuanced description but perhaps not much more clarity to the pattern.
-Geerat J. Vermeij

RESPONSE. We agree with reviewer 2 in that we did not aim at describing the processes behind the biotic transitions we found. While this is an exciting topic, we think that such exploration requires an in-depth study on the particular events that produced each transition, which is beyond the scope of a single manuscript.

For the first time, we have implemented a higher-order network framework of the metazoan macroevolution. We anticipate that the new framework will drive future research on emergent macroevolutionary structures at different scales.

Response to Referee 3

The manuscript titled A multiscale view of the Phanerozoic fossil record reveals the three major biotic transitions presents a reanalysis of a long-standing paradigm in Palaeontology, where the biota can be divided into 3 main evolutionary biotas. Furthermore, they present a novel suite of methods, at least for the field of palaeontology, which is multilayer network framework. This gives the manuscript two potentially citable aspects and a lot of interest that makes it suitable for publication in *Biology Communications*.

We thank William for the time he has put into reading our paper and for his useful suggestions that have helped us improve the manuscript.

I do, however, have some concerns and comments. At this stage, I am concerned at the high similarity to a recently published article by Drew Muscente et al. *PNAS* 115, 5217-5222 (2018) which has since received 21 citations [excl. self citations]. Both articles use the same dataset, both use

network analysis (even though the application here is more advanced), and both investigate changes in the evolutionary faunas. The results vary slightly with Muscente et al. recognizing 3-5 evolutionary faunas, and here only 3 and an earlier transition in the Cretaceous. I think others will also ask the same question; how does this new manuscript represent a significant advance?

RESPONSE. The reviewer is right that Muscente et al. (2018) deal with similar research questions and data. The difference is that they study pairwise interactions between components in a flattened single-layer network. Inevitably, representing this network of networks as a single aggregated network leads to information loss that can obscure the macroevolutionary pattern (see Domenico et al. 2015).

Our approach uses an input network that explicitly represents different time intervals and thus captures spatial (intra-layer) and temporal (across-layers) relationships in the data. By using multilayer networks with state nodes of geographic areas and taxa in each layer, we can delineate different types of communities: taxonomically overlapping, temporally overlapping, and entirely or partially overlapping. These overlapping assemblages provide more realistic representations of evolutionary faunas.

Besides, as the reviewer has noted, Muscente et al. used a distance-based approach to cluster their flattened network. It gives different solutions with a varying number of evolutionary faunas at different thresholds. How do we identify the best solution, the best macroevolutionary pattern? We have implemented a clustering algorithm that operates directly on the multilayer network and eliminates the subjectivity of the distance-based approaches.

REVISION. We have edited the section *A multilayer network representation of the Earth-Life System* that introduces the general idea of a multi-layer network representation and highlights differences with standard network approaches (Line 85).

REVISION. In addition, we have added a paragraph that summarizes the advantages of the multilayer network representation over standard approaches (Line 178):

In summary, our implementation of higher-order network models Edler et al. [2017] to understand the benthic marine fossil record provides a more reliable coarse-grained description of the macroevolutionary patterns in two ways: First, we provide a more realistic input network that explicitly represents different time intervals in Earth's history and better captures the relationship in the underlying paleontological data. Second, we employ a clustering approach that operates directly on the multilayer network, delineates taxonomically overlapping assemblages, and automatically defines the number of assemblages or evolutionary faunas, which eliminates the subjectivity of the distance-based approaches. As a result, the macroevolutionary pattern revealed using multilayer networks integrates two foundational hypotheses in paleobiology into a single story that gives a place to the Mesozoic Marine Revolution in the Phanerozoic marine macroevolution.

So, the follow up to that is, okay is this multilayer network framework the significant advance? After reading the manuscript I do not know, because I am struggling to understand what this multilayer approach is doing/ how it works and that is also a concern because I would argue I have a head start over most having used and published a network analysis application in Palaeontology (see Scientific Reports 10, 2176 (2020)).

REVISION. In the revised version, we have added a new figure (Figure S1) showing differences between standard and multilayer network representations. We used a small example because it allows us to visually represent all state nodes and layers of the multilayer network. This figure explains why representing this network of networks as a single aggregated network inevitably alters the trajectory of a random walker following the connections in the multilayer network, obscuring the macroevolutionary pattern (Supplementary Information):

Figure S1. A multilayer network visualization of fossil occurrence data and its single-layer flattened representation. **A.** Multilayer network at the stage-level resolution. **B.** Optimum network partition comprising three consecutive modules at the first hierarchical level. In this representation, grid cells are physical nodes restricted to a single layer. One state node in a specific layer represents a grid cell. Taxa are physical nodes that can occur in several layers. One state node per layer represents a taxon. In this visualization, taxa and grid cells are stacked. Each physical node occupies a particular location across all layers, which highlights that taxa gradually replace each other. To avoid clutter, we exclude inter-layer links between state nodes representing grid cells and taxa. **C.** Single-layer flattened representation of the multilayer network. **D.** Optimum network partition comprising four modules at the first hierarchical level. In this network, physical nodes represent taxa and grid cells independent of the stage(s) where they occur. **E.** Alluvial diagram comparing optimum solutions derived from applying Infomap on the multilayer network and its flattened representation. By simplifying this network of networks derived from stage-level resolution data (Fig. 2A) into an aggregated single-layer network, inevitably we change the trajectory of an entity following the connections of the original network and obscure the macroevolutionary pattern. Visualizations created using the platform MuxViz De Domenico et al. [2015a].

To summarise, I like this research and it is an outstanding achievement to understand and apply these complicated new methods and it should, therefore, be published but before that I think it needs some changes in how the ideas are communicated so it is more understandable and so the advances to the field are also more understandable.

RESPONSE. We acknowledge that it is a good idea to fully explain how our work compares against current published work on network analysis applications into palaeontology, including Muscente et al. 2018 and Scientific Reports 10, 2176 (2020). While the advantages multilayer network approaches have over standard network approaches, including those used in the papers cited by the reviewer, are well-known in network science, they are not in the paleontology community.

REVISION. In the revised version, we have added a paragraph highlighting the advantages of using multilayer networks compared to standard networks. Specifically, we highlight two aspects: (1) more accurate representations of the relationships in the underlying paleontological data, and (ii) a network clustering that can delineate complex communities that can only be observed in multilayer network representations. We have also included relevant references (Line 178):

In summary, our implementation of higher-order network models Edler et al. [2017] to understand the benthic marine fossil record provides a more reliable coarse-grained description of the macroevolutionary patterns in two ways: First, we provide a more realistic input network that explicitly represents different time intervals in Earth's history and better captures the relationship in the underlying paleontological data. Second, we employ a clustering approach that operates directly on the multilayer network, delineates taxonomically overlapping assemblages, and automatically defines the number of assemblages or evolutionary faunas, which eliminates the subjectivity of the distance-based approaches. As a result, the macroevolutionary pattern revealed using multilayer networks integrates two foundational hypotheses in paleobiology into a single story that gives a place to the Mesozoic Marine Revolution in the Phanerozoic marine macroevolution.

Also note, the dataset was not made available to me so fundamentally I could not review the data that underpins this study, and it appears that there was no vetting of the data. Stating the dataset used in this study is freely available at the Paleobiology Database is not acceptable because that database is always evolving, it is also standard practice to include the .csv file of the actual dataset that was used (see Muscente et al. as an example). Without including the data this study is not reproducible. Likewise, the code was not available so this study could not be reproduced I recommend using mybinder.org to make all this accessible and reproducible.

As noted above, to facilitate reproducibility in the revised version, we have:

- Described the PBDB download procedure in more detail (Materials and Methods).
- Included the PBDB filtered data (Data S1).
- Included the code to obtain bootstrapped networks from the original network.

In addition, we have indicated that the following resources are available online at the Infomap website www.mapequation.org:

- Source code for multilevel clustering with Infomap. It includes a client-side web application that makes it possible for users to run Infomap without any installation.
- Source code to perform the significance clustering.
- Source code to explore the landscape of solutions discussed in the appendix.
- The Alluvial Generator App we used to create Fig. 4.
- The Network Navigator App to create interactive zoomable maps for networks clustered with Infomap we used to generate Fig. 5.

Here are some more specific comments on the manuscript: John Phillips 1860 (not Jack Sepkoski) was the first to estimate Phanerozoic diversity and divide the curve into evolutionary faunas dubbed the Palaeozoic, Mesozoic and Cenozoic Life based on the British fossil record. Sepkoski and Raup much later (> 100 years) were some of the first, however, to quantify these changes and came up with slightly different results. This highlights a long standing concern I have about research conducted in the US, and that is that typically a lot of work done elsewhere in the world is overlooked or looked down at (personally I think this is because of the way palaeontologists network and the consequences of conferences). There are other examples of this, e.g., Line 179: Penn et al. is not an appropriate reference for the sentence. P.S., your results are very similar to Phillips (1860) which raises another point about statistical methods, where you should always use the simplest methods possible.

RESPONSE. The reviewer is right. Phillips (1860) was the first to estimate Phanerozoic diversity and divide this curve into three evolutionary faunas. However, Phillips (1860) estimated transitions only qualitatively and based on diversity loss associated with both end-Permian and end-Cretaceous mass extinction events. In contrast, we demonstrate the small effect that genus-level diversity losses from some global mass extinctions have on the evolutionary faunas. We have explained above how our method compares to standard methods.

Line 61-21 integrates the higher-order relationships over time this does not mean anything; please can you be more specific. As someone who incorporates machine learning into analysing data, I would think that you mean you are using all the metadata to find relationships, which is not correct. I could also interpret this to mean that you are looking at different taxonomic levels, which is also not correct.

RESPONSE. We agree that we must remove all ambiguity by explaining how our work compares against published work on network analysis applied to palaeontology, including Muscente et al 2018 and Scientific Reports 10, 2176 (2020).

REVISION We have edited the caption of figure 1 to include the main differences between our approach and current approaches:

Figure 1. Network models used in macroevolution. A-B. Standard first-order network representations. A. Bipartite occurrence network. This representation comprises two sets of physical nodes that represent geographic areas and taxa Rojas et al. /2017]. B. Unipartite co-occurrence networks Muscente et al. /2018], Kocsis et al. /2018]. These representations are weighted projections of the bipartite network onto each set of physical nodes. C. Higher-order multilayer representation of temporal data. In this network, distinct state nodes represent a physical node in each layer where the physical node occurs Edler et al. /2017]. The trajectory of a random walker guided by the links between the nodes models network flows (30). Network flows are of first-order when random walker movements are constrained to single layers and of higher-order when they can move within and between layers (25). The aggregation process that simplifies a multilayer network into a single-layer representation alters the network flows, obscuring the modular structures.

REVISION: We have now explained the differences in detail in the section A multilayer network representation of the Earth-Life System (Line 85).

As I understand the layers represent different time intervals, which would stop 2 nodes connecting if theyre separated by a significant number of layers if that is the case, how do you end up with fewer evolutionary faunas then using a bipartite or unipartite method because this approach should be splitting the data more often?

RESPONSE: The information lost in the aggregation from multilayer to one-layer representation decides if and how the number of evolutionary faunas changes. The number can go up or down or stay the same. In

the revised version, we provide an example showing that the number of modules obtained from clustering the single-layer flattened network could be higher than those obtained from clustering a multilayer network representing temporal data (Figure S4).

REVISION: We added a paragraph explaining differences between our approach and standard approaches (Line: 144)

Aggregation procedures that simplify a multilayer network representation of the Earth-Life System into a single-layer representation, such as those traditionally used in macroevolutionary research Muscente et al. [2018], Foster et al. [2020], change the trajectory of a random walker following the connections in the network, and inevitably obscure the macroevolutionary pattern (Fig. S4).

REVISION: We also added this paragraph highlighting how the flattening alters the trajectory of a random walker on the network and thus the delineated macroevolutionary pattern (Line 106):

Multilayer network representations that consider higher-order dependencies in the row data Xu et al. [2016], Edler et al. [2017], such as temporal relationships inherent to the fossil record known as the principle of faunal succession, have been shown to reveal community structures that standard models based on pairwise interactions cannot capture Lambiotte et al. [2019]. Clustering methods developed for standard networks cannot reliably identify overlapping modular structures Magnani et al. [2020]. Without the concept of layers, in the traditional network approaches used in macroevolutionary research, a random walker visiting a taxon will move at the same rate to grid cells from different ages because aggregating layers washes out temporal information (Fig. ??A-B). To overcome this limitation, we implemented a higher-order framework that explicitly represents different time intervals in Earth's history. In this network representation, a random walker visiting a physical node or taxon is located in a particular layer (t_0) at a state node and will move at different rates to grid cells from the same layer and neighbouring layers (Fig. ??C) (see Network analysis). In this way, we captures cross layers structures representing evolutionary faunas without destroying intra-layer dynamics in the underlying Earth-Life System.

You also explain the geographic range is used in the analysis, although I do not understand how, but this would then make the dataset prone to many sampling biases. I know you acknowledge this, but how does a parametric bootstrap method fix a lack of data/ incorrect data in the Paleobiology Database. You do not say in the actual article that you vetted the data. Furthermore, why does geography matter when studying evolutionary faunas? Does it matter that a family of snails that evolved in the tropics can now only be found in the depths of Antarctica when understanding evolutionary faunas?

RESPONSE. We did not intend that interpretation. We plotted heatmaps of genus richness across time only to show where the data comes from in terms of latitudinal richness. We focus exclusively on the temporal patterns in our paper. (see new Figure 3).

REVISION. We have edited the figure caption to highlight the purpose of the heatmaps of genus richness across time.

References

Terms like physical nodes state nodes need explaining in a way an undergraduate would understand.

REVISION. We have now explained the concept of state nodes when introducing the multilayer network (Line 124):

Our multilayer network representation of the Earth-Life System treats ordered geological stages in the standard time scale Gradstein et al. [2004] as layers assembled into a multilayer network Mucha et al. [2010], De Domenico et al. [2015b], Xu et al. [2016]. In this multilayer network (Data S1), physical nodes, which in conventional models represent Phanerozoic benthic marine taxa Peters and McClennen [2016] and geographic areas where they have been described, are divided into state nodes, which represent temporal relationships in the Earth-Life System (Fig. ??C). Specifically, for each taxon, we create one state node per layer where the taxon occurs. In this way, the trajectory of a random walker following the connections between

state nodes in the stage-level fossil data multilayer network captures higher-order temporal dependencies between the physical nodes.

REVISION. We also explained this concept in figure 1.

Lines 103-109. It is better to use well-known stages, right?
Like Maastrichtian.

REVISION. We have edited the section *A multilayer network representation of the Earth-Life System* to explain our network input. We introduced the concept of network flows to explain the differences between clustering standard and multilayer networks better (Line 136):

In this way, the trajectory of a random walker following the connections between state nodes in the stage-level fossil data multilayer network captures higher-order temporal dependencies between the physical nodes. The trajectory of a random walker guided by the links between the nodes models network flows Rosvall and Bergstrom [2008]. Network flows are of first-order when random walker movements are constrained to individual layers and of higher-order when they can move within and between layers Edler et al. [2017]. Aggregation procedures that simplify a multilayer network representation of the Earth-Life System into a single-layer representation, such as those traditionally used in macroevolutionary research Muscente et al. [2018], Foster et al. [2020], change the trajectory of a random walker following the connections in the network, and inevitably obscure the macroevolutionary pattern (Fig. S4).

RESPONSE. We have also created Figure S5 to explain the differences.

Lines 125-143 do not explain what you are doing in a way that can be repeated without extensive additional work and reading. You do not explain what data you are using, okay you say Paleobiology Database, but you also say taxa, is this species-level, genus-level, family-level, order-level? One aspect I like in Muscente et al. is that they run their analysis at all these levels to see the differences in the results.

Finally, I looked at ref 24 which you cite a lot. I am now asking the question why didnt you use a multilayer memory network? This would have been better adapted to handling sampling problems, especially problems like Lazarus taxa if I understood it right.

RESPONSE. We use multilayer memory networks. Our multilayer network is a particular case of those memory networks cited in Edler et al. (2017). Sparse memory networks is a general framework that uses state nodes to describe higher-order dynamics and physical nodes to represent the physical objects. We think this point is clear now with the new explanations of the method.

Lines 155-158 differs from standard geological eras (Adjusted Mutual Information, AMI = 0.71), indicating that not all major biotic transitions occur at their boundaries. This is very hard to believe, biostratigraphy is based on major turnovers in fauna and accordingly this analysis did not [some would argue failed to] recognise the turnover at the Mesozoic/Cenozoic boundary. Since I could not see the data I could not go into further detail as to why this is the case, but isnt it strange that ammonites (I am using an extreme example) are clustering with Cenozoic genera? The issue is probably that you are only looking at invertebrates, if you included the chondrichthyans I am sure you would see a Cenozoic cluster appear. Another way to look at it, is does this multilayer method really work? If you did the bipartite and unipartite methods do you get different results? My paper I mentioned before, Scientific Reports 10, 2176 (2020), did find a significant turnover in benthic invertebrates at the K/Pg boundary. It is probably then a good idea to compare this new method with the old network methods you introduce earlier on. P.S., my issue with network analyses is that it seems to under cluster assemblages and this new approach does not appear to be immune to this problem.

RESPONSE. We used the core of the benthic marine invertebrates that have traditionally been used to access this macroevolutionary pattern (including Muscente et al.). For completeness, we indicated in the manuscript that patterns would change with input data. Yes, the Scientific Reports 10, 2176 (2020) did find a significant turnover in benthic invertebrates at the K/Pg boundary, and we found that at lower hierarchical levels. However, our approach that provides the optimal hierarchical solution shows that genus richness loss at the K/Pg boundary does not result in a transition between evolutionary faunas. This result is consistent with Sepkoski's seminal work.

Infomap, the network clustering approach we use in the analysis, is known to not underfit networks with planted or known clusters (A. Lancichinetti and S. Fortunato *Community detection algorithms: a comparative analysis*, Physical Review E 80:056117 (2009)). Working with limited data and higher-order networks that can represent high-dimensional spaces, the risk is instead to overfit. That is why we employ the significance analysis. Other studies have compared distance-based clustering approaches with network methods and concluded that the latter are more sensitive (N.J. Bloomfield, N. Knerr, and F. Encinas-Viso, *A comparison of network and clustering methods to detect biogeographical regions*, Ecography 41:1–10 (2018)).

REVISION. Our exploration of the landscape of solutions showed the small effects of the K/Pg mass extinction event on the temporal structuring of the evolutionary faunas (Figure 4).

Line 178–179: neither ref. 34 nor 35 are appropriate. You need citations that show evolutionary changes and comparisons of mass extinction events, these two papers do not do that and gives the appearance of a lack of understanding. I highlighted these ones because I know them best, so please check the quality of the referencing throughout.

REVISION. We thank the reviewer for the suggestions. We have added the following references suggested by reviewer 1 and 2:

Dunhill AM, Foster WJ, Sciberras J, and Twitchett RJ. 2018. Impact of the Late Triassic mass extinction on functional diversity and composition

of marine ecosystems. Palaeontology 61:133-148.

Stanley SM (2016) Estimates of the magnitudes of major marine mass extinctions in earth559history. Proceedings of the National Academy of Sciences 113(42):E6325–E6334

Figure 5 is not cited in the manuscript.

REVISION. We thank the reviewer for spotting the mistake. We have revised the manuscript and cited all figures.

I do not understand how Figs. 4-5 contribute to the article when you do not discuss these lower order structures.

RESPONSE. We plotted modules in lower hierarchical levels to show that our approach, in contrast to other methods, captures temporal (inter-layer) dynamics without destroying spatial (intra-layer) relationships. Observing meaningful bioregions suggest that including both types of relationships is essential. For instance, Kocis et al. (2018) used a standard network approach to map those bioregions. Their approach did not allow them to delineate evolutionary faunas because they ran Infomap on a two-mode projected network. We believe these results favour the multilayer approach we introduce to macroevolutionary research. We can map more Phanerozoic bioregions but it exceeds the scope of our work.

REVISION. We have linked the figures to our previous discussion on the advantages of multilayer networks. Now we cite all figures in the manuscript. Also, we have edited the figure caption to describe the relevance of the figure better:

Figure 6. Examples of evolutionary bioregions recovered at lower hierarchical levels. These modular structures form geographically coherent units that change through time and underlie the global marine mega-assemblages. Circles represent the centre of the geographic cells coloured by their module affiliation (see Data S2). By recovering bioregions at lower hierarchical levels, we demonstrate that our multilayer approach captures temporal (inter-stage) relationships without destroying spatial (intra-stage) dynamics.

I do not understand Fig. 3 and why some of the bootstraps are highlighted as Sepkoski's evolutionary faunas. How would I know from the figure? What do the white gaps mean?

REVISION. We have improved the figure caption, which now explains how to read the figure. We also re-labelled the figure to make it clearer.

Figure 4. The solution landscape. Alluvial diagram comparing the four-phase reference solution against alternative solutions obtained from bootstrapped networks. Vertical stacks connected by streamlines represent the networks. In each stack, a block represents nodes clustered in a module or mega-assemblage. Streamlines connect modules that contain the same nodes. The height of the streamline is proportional to the number of shared nodes between the connected modules. Splitting or merging streamlines indicate biotic transitions. The bootstrap strategy demonstrate high robustness of the four mega-assemblages in the reference solution (Table S1), supporting the global biotic transitions delineated at the end-Cambrian, end-Permian, and mid-Cretaceous. The bootstrapped solutions also capture the small effect the Cretaceous-Paleogene extinction event (K-Pg) has on the mid-Cretaceous to Quaternary mega-assemblage (mKr-Q), and the instability of the global biosphere through the Triassic-Jurassic (Tr-Jr) extinction event (see Fig. 2). Abbreviations: Cambrian (Cm), Paleozoic (Pz), Triassic to lower Cretaceous (Tr-lKr), and mid-Cretaceous to Quaternary (mKr-Q) mega-assemblages.

The formatting of the article was slightly off, means words had spaces within them making it harder to read. E.g., line 87 n etwork i ncluding. E.g., line 99.

REVISION. We apologise for those mistakes. We have revised the whole manuscript and proofread it carefully.

Lines 14-22, these sentences are real tongue twisters and hard to read.

REVISION. We have revised the sentence to make it easier for the readers.

That written quality of the article was high!
I hope you have found my review constructive and I am available if you have any further questions,

RESPONSE. Again, we thank the reviewer for the time and effort invested in reviewing our manuscript.

References

Andrew B. Smith. Marine diversity through the Phanerozoic: problems and prospects. *Journal of the Geological Society*, 164(4):731–745, July 2007. ISSN 0016-7649, 2041-479X. doi: 10.1144/001676492006-184. URL <http://jgs.lyellcollection.org/lookup/doi/10.1144/0016/76492006-184>.

Joaquín Calatayud, Ruben Bernardo-Madrid, Magnus Neuman, Alexis Rojas, and Martin Rosvall. Exploring the solution landscape enables more reliable network community detection. *Physical Review E*, 100(5):052308, November 2019. ISSN 2470-0045, 2470-0053. doi: 10.1103/PhysRevE.100.052308. URL <https://link.aps.org/doi/10.1103/PhysRevE.100.052308>.

- Zhong-Qiang Chen and Michael J. Benton. The timing and pattern of biotic recovery following the end-Permian mass extinction. *Nature Geo-science*, 5(6):375–383, June 2012. ISSN 1752-0894, 1752-0908. doi: 10.1038/ngeo1475. URL <http://www.nature.com/articles/ngeo1475>.
- Haijun Song, Paul B. Wignall, and Alexander M. Dunhill. Decoupled taxonomic and ecological recoveries from the Permo-Triassic extinction. *Science Advances*, 4(10):eaat5091, October 2018. ISSN 2375-2548. doi: 10.1126/sciadv.aat5091. URL <http://advances.sciencemag.org/lookup/doi/10.1126/sciadv.aat5091>.
- Alexander M. Dunhill, William J. Foster, James Sciberras, and Richard J. Twitchett. Impact of the Late Triassic mass extinction on functional diversity and composition of marine ecosystems. *Palaeontology*, 61(1): 133–148, January 2018. ISSN 00310239. doi: 10.1111/pala.12332. URL <http://doi.wiley.com/10.1111/pala.12332>.
- M. Hautmann, F. Stiller, C. Huawei, and S. Jingeng. Extinction-Recovery Pattern of Level-Bottom Faunas Across the Triassic-Jurassic Boundary in Tibet: Implications for Potential Killing Mechanisms. *PALAIOS*, 23(10):711–718, October 2008. ISSN 0883-1351. doi: 10.2110/palo.2008.p08-005r. URL <https://pubs.geoscienceworld.org/palaios/article/23/10/711-718/145951>.
- Steven M. Stanley. Estimates of the magnitudes of major marine mass extinctions in earth history. *Proceedings of the National Academy of Sciences*, 113(42):E6325–E6334, October 2016. ISSN 0027-8424, 1091-6490. doi: 10.1073/pnas.1613094113. URL <http://www.pnas.org/lookup/doi/10.1073/pnas.1613094113>.
- Daniel Edler, Ludvig Bohlin, and and Rosvall. Mapping Higher-Order Network Flows in Memory and Multilayer Networks with Infomap. *Algorithms*, 10(4):112, September 2017. ISSN 1999-4893. doi: 10.3390/a10040112. URL <http://www.mdpi.com/1999-4893/10/4/112>.
- M. De Domenico, M. A. Porter, and A. Arenas. MuxViz: a tool for multilayer analysis and visualization of networks. *Journal of Complex Networks*, 3(2):159–176, June 2015a. ISSN 2051-1310, 2051-1329. doi:

10.1093/comnet/cnu038. URL <https://academic.oup.com/comnet/article-lookup/doi/10.1093/comnet/cnu038>.

Alexis Rojas, Pedro Patarroyo, Liang Mao, Peter Bengtson, and Michał Kowalewski. Global biogeography of Albian ammonoids: A network-based approach. *Geology*, 45(7):659–662, July 2017. ISSN 0091-7613, 1943-2682. doi: 10.1130/G38944.1. URL <https://pubs.geoscienceworld.org/geology/article/45/7/659-662/207876>.

A. D. Muscente, Anirudh Prabhu, Hao Zhong, Ahmed Eleish, Michael B. Meyer, Peter Fox, Robert M. Hazen, and Andrew H. Knoll. Quantifying ecological impacts of mass extinctions with network analysis of fossil communities. *Proceedings of the National Academy of Sciences*, 115(20): 5217–5222, May 2018. ISSN 0027-8424, 1091-6490. doi: 10.1073/pnas.1719976115. URL <http://www.pnas.org/lookup/doi/10.1073/pnas.1719976115>.

Ádám T. Kocsis, Carl J. Reddin, and Wolfgang Kiessling. The biogeographical imprint of mass extinctions. *Proceedings of the Royal Society B: Biological Sciences*, 285(1878):20180232, May 2018. ISSN 0962-8452, 1471-2954. doi: 10.1098/rspb.2018.0232. URL <http://rspb.royalsocietypublishing.org/lookup/doi/10.1098/rspb.2018.0232>.

William J. Foster, Christopher L. Garvie, Anna M. Weiss, A. D. Muscente, Martin Aberhan, John W. Counts, and Rowan C. Martindale. Resilience of marine invertebrate communities during the early Cenozoic hyper-thermals. *Scientific Reports*, 10(1):2176, December 2020. ISSN 20452322. doi: 10.1038/s41598-020-58986-5. URL <http://www.nature.com/articles/s41598-020-58986-5>.

Jian Xu, Thanuka L. Wickramaratne, and Nitesh V. Chawla. Representing higher-order dependencies in networks. *Science Advances*, 2(5):e1600028, May 2016. ISSN 2375-2548. doi: 10.1126/sciadv.1600028. URL <https://advances.sciencemag.org/lookup/doi/10.1126/sciadv.1600028>.

Renaud Lambiotte, Martin Rosvall, and Ingo Scholtes. From networks to optimal higher-order models of complex systems. *Nature*

- Physics*, 15(4):313–320, April 2019. ISSN 1745-2473, 1745-2481. doi: 10.1038/s41567-019-0459-y. URL <http://www.nature.com/articles/s41567-019-0459-y>.
- Matteo Magnani, Obaida Hanteer, Roberto Interdonato, Luca Rossi, and Andrea Tagarelli. Community Detection in Multiplex Networks. *arXiv:1910.07646 [physics]*, August 2020. URL <http://arxiv.org/abs/1910.07646>. arXiv: 1910.07646.
- F. M. Gradstein, James G. Ogg, and A. Gilbert Smith. *A geologic time scale 2004*. Cambridge University Press, Cambridge, UK ; New York, 2004. ISBN 978-0-521-78142-8 978-0-521-78673-7.
- Peter J Mucha, Thomas Richardson, Kevin Macon, Mason A Porter, and Jukka-Pekka Onnela. Community structure in time-dependent, multi-scale, and multiplex networks. *Science*, 328(5980):876–878, 2010.
- Manlio De Domenico, Andrea Lancichinetti, Alex Arenas, and Martin Ros-vall. Identifying modular flows on multilayer networks reveals highly overlapping organization in interconnected systems. *Physical Review X*, 5(1):011027, 2015b.
- Shanan E. Peters and Michael McClellan. The Paleobiology Database application programming interface. *Paleobiology*, 42(01):1–7, February 2016. ISSN 0094-8373, 1938-5331. doi: 10.1017/pab.2015.39. URL http://www.journals.cambridge.org/abstract_S0094837315000391.
- M. Rosvall and C. T. Bergstrom. Maps of random walks on complex networks reveal community structure. *Proceedings of the National Academy of Sciences*, 105(4):1118–1123, January 2008. ISSN 0027-8424, 10916490. doi: 10.1073/pnas.0706851105. URL <http://www.pnas.org/cgi/doi/10.1073/pnas.0706851105>.

Reviewers' comments:

Reviewer #2 (Remarks to the Author):

I think the improvements are adequate even if this paper still remains an abstract description. I would have put Figure S1 in the main body of the paper.

Reviewer #3 (Remarks to the Author):

I think the revisions made by the authors are appropriate and clarify most of my comments in the last review, and I will reiterate that I think this article represents an important contribution to the discipline and I am very positive about the article! Just a few quick points that I think still need addressing:

Still this article attributes the macroevolutionary faunas to Jack Sepkoski and that they are testing this hypothesis. I still don't like this, so let me explain why in another way: This hypothesis was already falsified by Muscente et al. (and most likely by John Alroy's work too, but I don't remember him referring to it), so by not acknowledging that the macroevolutionary faunas recognized by Muscente et al. you are permitting readers to do the same with your article, i.e., why should we follow your article's findings if you're not following Muscente et als. That is why I think you should drop referring to the macroevolutionary faunas as Jack Sepkoski's.

In addition, it is now clear that your macroevolutionary faunas record a signature of the end-Cretaceous mass extinction, but you don't acknowledge this beyond the figure, and I get it they are not in the final network output but the network analysis is recognizing it as significant. So again, your results are actually closely related to the biostrat, i.e., the original macroevolutionary faunas of Phillips. I encourage you to acknowledge this point – especially as it's not surprising given the magnitude of the end-Cretaceous event.

The authors attribute the macroevolutionary faunas to the end-Cambrian and end-Permian extinctions, but actually it is about the subsequent radiations that causes the change in evolution, i.e., the Great Ordovician Biodiversification Event (GOBE), the post-Permian radiation, and the Mesozoic Marine Revolution (MMR). The extinctions just reset diversity, it is what happens afterwards that governs the evolutionary trajectory. Unless you want to refer to extinction selectivity and how that ties to the subsequent radiations, but this is still different to what you're describing.

What is the Earth-Life system? Do you mean the Biosphere? I searched the term in textbooks and couldn't get any clarification – except one suggestion that it was about the coevolution of life and the planet, but your article is just the about the evolution of life. I would change this term to something else.

Fig. 3 was half off the page, so I didn't really see the whole figure.

Line 89: change biotic to biota

Line 156: what does "washes out" mean?

Line 164: change captures to capture

Line 264: you have cited an end-Triassic paper for the end-Permian extinction . . . either see comment

below about ref 48 or you can cite Foster & Twitchett (2014) *Nature Geoscience* 7, 233-238. Actually, a better reference would be the earlier work of Doug Erwin on the end-Permian, but you seem to have a preference to the more recent literature.

Line 352: delete Earth's largest. This is subjective, some may argue that the Cryogenian, end-Ediacaran or end-Cambrian is the largest.

Line 382: it is clear that the end-Cretaceous mass extinction is picked up in the network analysis, even if not in the final output.

Change ref 48 (Penn et al.). To suggest what is difficult, there is actually very little work on the P/Tr turnover. I would suggest referencing Twitchett and Foster (2012) *Post-Permian radiation*. *Els*. But this is an encyclopaedia article, and a new version is being published in 1-2 weeks, so I would otherwise suggest Payne and Clapham (2012) *Annual Review of Earth and Planetary Sciences* 40, 89-111. Again, if it is actually the extinction selectivity that you want to cite then see <https://www.biorxiv.org/content/10.1101/2020.10.09.332999v1>

Point-by-point response to the referees' comments

Response to Referee 2

I think the improvements are adequate even if this paper still remains an abstract description. I would have put Figure S1 in the main body of the paper.

RESPONSE. We are thankful for the reviewer's comment on the revised version. However, we did not add Figure S1 into the main body of the paper because its purpose is only to visually show that clustering a multilayer network provides different modular structures compared with those derived from its flattened representation. The current version of the manuscript addresses this point in lines 156–162.

Response to Referee 3

I think the revisions made by the authors are appropriate and clarify most of my comments in the last review, and I will reiterate that I think this article represents an important contribution to the discipline and I am very positive about

the article! Just a few quick points that I think still need addressing:

Still this article attributes the macroevolutionary faunas to Jack Sepkoski and that they are testing this hypothesis. I still dont like this, so let me explain why in another way: This hypothesis was already falsified by Muscente et al. (and most likely by John Alroys work too, but I dont remember him referring to it), so by not acknowledging that the macroevolutionary faunas recognized by Muscente et al. you are permitting readers to do the same with your article, i.e., why should we follow your articles findings if youre not following Muscente et als. That is why I think you should drop referring to the macroevolutionary faunas as Jack Sepkoskis.

RESPONSE. We are happy to read that the reviewer thinks our article represents an important contribution to the discipline and that our revisions have clarified most comments.

We agree with the reviewer that we should better acknowledge the recent studies that have questioned the validity of Sepkoski's fauna. Nevertheless, Sepkoski's evolutionary faunas remain a fundamental hypothesis in numerous macroevolutionary papers published in the last few years. For example, the following recent papers used Jack Sepkoski's Cambrian, Paleozoic or modern faunas as underlying framework and assumed they are still well-established macroevolutionary units:

- Brayard et al. [2017]
- Dineen et al. [2019]
- Wolfe et al. [2019]
- Rong et al. [2020]
- Fan et al. [2020]
- Cribb and Bottjer [2020]

REVISION. We have revised the introduction's opening paragraph to highlight the study by Muscante et al. and to emphasize that Sepkoski's hypothesis remains an important conceptual driver of macroevolutionary research.

However, while the hypothesis continues to serve as a conceptual platform for many recent studies Brayard et al. [2017], Dineen et al. [2019], Fan et al. [2020], some analyses question the validity of Sepkoski's hypothesis Muscante et al. [2019]. [Lines: 12–15]

In addition, it is now clear that your macroevolutionary faunas record a signature of the end-Cretaceous mass extinction, but you don't acknowledge this beyond the figure, and I get it they are not in the final network output but the network analysis is recognizing it as significant. So again, your results are actually closely related to the biostrat, i.e., the original macroevolutionary faunas of Phillips. I encourage you to acknowledge this point especially as it's not surprising given the magnitude of the end-Cretaceous event.

RESPONSE. The reviewer is right that our analysis captures the effects of the end-Cretaceous mass extinction event on the marine faunas. Our multiscale approach also shows that such an event does not drive the first hierarchical level's modular structures.

REVISION. In the revised version, we have added a sentence indicating how the end-Cretaceous mass extinction event affects the modular structure of the assembled network:

Our multiscale analysis shows that the end-Cretaceous mass extinction event controls the emergence of nested modules at the second hierarchical level representing marine sub-faunas. This global event does not affect the large-scale modular structure comprising four marine mega-assemblages (Table S1). [Lines: 277–280]

In addition, we have added the 19th-century reference as suggested by the

reviewer:

Phillips [1860] provided the first global assessment based on a qualitative approach
[Lines: 24–25]

The authors attribute the macroevolutionary faunas to the end-Cambrian and end-Permian extinctions, but actually it is about the subsequent radiations that causes the change in evolution, i.e., the Great Ordovician Biodiversification Event (GOBE), the post-Permian radiation, and the Mesozoic Marine Revolution (MMR). The extinctions just reset diversity, it is what happens afterwards that governs the evolutionary trajectory. Unless you want to refer to extinction selectivity and how that ties to the subsequent radiations, but this is still different to what youre describing.

RESPONSE. We did not intend to suggest that the extinction events caused all subsequent evolutionary history of marine biotas. Instead, we view those events as triggering events that paved the way toward establishing new evolutionary faunas.

REVISION. We have reviewed the text and made appropriate edits throughout to ensure that we are not postulating that extinctions events governed the subsequent evolutionary history but only indicate that such extinctions initiated major biotic transitions:

Therefore, the first two major biotic transitions delineated via multilayer network analysis were more likely triggered by mass extinction events. However, those global events did not govern the subsequent evolutionary history of the marine biotas. [Lines: 240–244]

A mass extinction event initiated the biotic transition at the end-Permian, but more studies are required to understand the subsequent evolutionary history of the marine mega-assemblages. [Lines: 312–315]

We also replaced the following sentence: "For instance, we show that both long-term ecological interactions and global geological perturbations seem

to have played a critical role in shaping the large-scale structure of the marine animals." [Lines: 345–348] by the following:

For instance, we show that both long-term ecological interactions and global geological perturbations seem to have played critical roles in shaping the mega-assemblages that dominated Phanerozoic oceans. [Lines: 352-355]

What is the Earth-Life system? Do you mean the Biosphere? I searched the term in textbooks and couldn't get any clarification except one suggestion that it was about the coevolution of life and the planet, but your article is just about the evolution of life. I would change this term to something else.

RESPONSE. Without a proper explanation, we understand the reviewer's concerns about the name. We call the study system "the Earth-Life system" because we use a bipartite network with two types of nodes representing animals and the Earth's regions where they occurred (grid cells).

REVISION. We have added a sentence explaining the choice of name.

Because the assembled network comprises nodes representing both animals and areas where they occur, we call our study system the Earth-Life system. Although we focus on the set of nodes representing animals, the bipartite network can also reveal spatio-temporal patterns in the distribution of both life and rocks. [Lines: 131–136]

Fig. 3 was half off the page, so I didn't really see the whole figure.

REVISION. We have revised the format.

Line 89: change biotic to biota

REVISION. The word "transitions" was missing. The sentence "major biotic in Earth's history" was replaced by "major biotic transitions in Earth's history". [Line: 78]

Line 156: what does washes out mean?

RESPONSE. We mean distorts information.

REVISION. We have replaced "washes out" by "distorts" [Lines: 106 and 122].

Line 164: change captures to capture. [line: 126]

REVISION. Corrected as suggested.

Line 264: you have cited an end-Triassic paper for the end-Permian extinction . . . either see comment below about ref 48 or you can cite Foster & Twitchett (2014) Nature Geoscience 7, 233-238. Actually, a better reference would be the earlier work of Doug Erwin on the end-Permian, but you seem to have a preference to the more recent literature.

REVISION. The reviewer is correct. We have now cited Foster Twitchett (2014) as suggested.

Line 352: delete Earths largest. This is subjective, some may argue that the Cryogenian, end-Ediacaran or end-Cambrian is the largest.

REVISION. Although we believe the end-Permian mass extinction is widely considered as the Earth's largest extinction event, we have changed the "Earth's largest" to "the most severe biotic crisis of the past 500 million years" [Lines:239-240]

Change ref 48 (Penn et al.). To suggest what is difficult, there is actually very little work on the P/Tr turnover. I would suggest referencing Twitchett and Foster (2012)

Post-Permian radiation. Els. But this is an encyclopaedia article, and a new version is being published in 1-2 weeks, so I would otherwise suggest Payne and Clapham (2012) Annual Review of Earth and Planetary Sciences 40, 89-111. Again, if it is actually the extinction selectivity that you want to cite then see

<https://www.biorxiv.org/content/10.1101/2020.10.09.332999v1>

REVISION. We replaced the Penn et al. reference by Twitchett and Foster (2012) as suggested.

References

- Arnaud Brayard, L. J. Krumenacker, Joseph P. Botting, James F. Jenks, Kevin G. Bylund, Emmanuel Fara, Emmanuelle Vennin, Nicolas Olivier, Nicolas Goudemand, Thomas Saucède, Sylvain Charbonnier, Carlo Romano, Larisa Doguzhaeva, Ben Thuy, Michael Hautmann, Daniel A. Stephen, Christophe Thomazo, and Gilles Escarguel. Unexpected Early Triassic marine ecosystem and the rise of the Modern evolutionary fauna. *Science Advances*, 3(2):e1602159, February 2017. ISSN 2375-2548. doi: 10.1126/sciadv.1602159. URL <https://advances.sciencemag.org/lookup/doi/10.1126/sciadv.1602159>.
- Ashley A. Dineen, Peter D. Roopnarine, and Margaret L. Fraiser. Ecological continuity and transformation after the Permo-Triassic mass extinction in northeastern Panthalassa. *Biology Letters*, 15(3):20180902, March 2019. ISSN 1744-9561, 1744-957X. doi: 10.1098/rsbl.2018.0902. URL <https://royalsocietypublishing.org/doi/10.1098/rsbl.2018.0902>.
- Joanna M. Wolfe, Jesse W. Breinholt, Keith A. Crandall, Alan R. Lemmon, Emily Moriarty Lemmon, Laura E. Timm, Mark E. Siddall, and Heather D. Bracken-Grissom. A phylogenomic framework, evolutionary timeline and genomic resources for comparative studies of deca-pod crustaceans. *Proceedings of the Royal Society B: Biological Sciences*, 286(1901):20190079, April 2019. ISSN 0962-8452, 1471-2954.

doi: 10.1098/rspb.2019.0079. URL <https://royalsocietypublishing.org/doi/10.1098/rspb.2019.0079>.

Jiayu Rong, D.A.T. Harper, Bing Huang, Rongyu Li, Xiaole Zhang, and Di Chen. The latest Ordovician Himantian brachiopod faunas: New global insights. *Earth-Science Reviews*, 208:103280, September 2020. ISSN 00128252. doi: 10.1016/j.earscirev.2020.103280. URL <https://linkinghub.elsevier.com/retrieve/pii/S0012825220303263>.

Jun-xuan Fan, Shu-zhong Shen, Douglas H. Erwin, Peter M. Sadler, Norman MacLeod, Qiu-ming Cheng, Xu-dong Hou, Jiao Yang, Xiang-dong Wang, Yue Wang, Hua Zhang, Xu Chen, Guo-xiang Li, Yi-chun Zhang, Yu-kun Shi, Dong-xun Yuan, Qing Chen, Lin-na Zhang, Chao Li, and Ying-ying Zhao. A high-resolution summary of Cambrian to Early Triassic marine invertebrate biodiversity. *Science*, 367(6475):272–277, January 2020. ISSN 0036-8075, 1095-9203. doi: 10.1126/science.aax4953. URL <https://www.sciencemag.org/lookup/doi/10.1126/science.aax4953>.

Alison T. Cribb and David J. Bottjer. Complex marine bioturbation ecosystem engineering behaviors persisted in the wake of the end-Permian mass extinction. *Scientific Reports*, 10(1):203, December 2020. ISSN 2045-2322. doi: 10.1038/s41598-019-56740-0. URL <http://www.nature.com/articles/s41598-019-56740-0>.

A. D. Muscente, Natalia Bykova, Thomas H. Boag, Luis A. Buatois, M. Gabriela Mángano, Ahmed Eleish, Anirudh Prabhu, Feifei Pan, Michael B. Meyer, James D. Schiffbauer, Peter Fox, Robert M. Hazen, and Andrew H. Knoll. Ediacaran biozones identified with network analysis provide evidence for pulsed extinctions of early complex life. *Nature Communications*, 10(1):911, December 2019. ISSN 2041-1723. doi: 10.1038/s41467-019-08837-3. URL <http://www.nature.com/articles/s41467-019-08837-3>.

John Phillips. *Life on the earth its origin and succession* / by John Phillips. Macmillan, Cambridge [Eng] ; 1860. doi: 10.5962/bhl.title.22153. URL <http://www.biodiversitylibrary.org/bibliography/22153>.